# Sample Selection with Uncertainty of Losses for Learning with Noisy Labels

**Xiaobo Xia**[1]    **Tongliang Liu**[1†]   **Bo Han**[2]
**Mingming Gong**[3]    **Jun Yu**[4]    **Gang Niu**[5]    **Masashi Sugiyama**[5,6]
[1]TML Lab, The University of Sydney    [2]Hong Kong Baptist University
[3]The University of Melbourne    [4]University of Science and Technology of China
[5]RIKEN AIP    [6]The University of Tokyo

## Abstract

In learning with noisy labels, the *sample selection* approach is very popular, which regards *small-loss* data as correctly labeled during training. However, losses are generated on-the-fly based on the model being trained with noisy labels, and thus *large-loss* data are *likely but not certain* to be incorrect. There are actually two possibilities of a large-loss data point: (a) it is mislabeled, and then its loss *decreases slower* than other data, since deep neural networks "learn patterns first"; (b) it belongs to an underrepresented group of data and *has not been selected yet*. In this paper, we incorporate the uncertainty of losses by adopting *interval estimation* instead of *point estimation* of losses, where lower bounds of the *confidence intervals* of losses derived from *distribution-free concentration inequalities*, but not losses themselves, are used for sample selection. In this way, we also give large-loss but less selected data a try; then, we can better distinguish between the cases (a) and (b) by seeing if the losses *effectively decrease* with the uncertainty after the try. As a result, we can better explore underrepresented data that are correctly labeled but seem to be mislabeled *at first glance*. Experiments demonstrate that the proposed method is superior to baselines and robust to a broad range of label noise types.

## 1 Introduction

Learning with noisy labels is one of the most challenging problems in weakly-supervised learning, since noisy labels are ubiquitous in the real world (Mirzasoleiman et al., 2020; Yu et al., 2019; Nishi et al., 2021; Arazo et al., 2019; Yang et al., 2021a; Bai & Liu, 2021). For instance, both crowdsourcing and web crawling yield large numbers of noisy labels everyday (Han et al., 2018). Noisy labels can severely impair the performance of deep neural networks with strong memorization capacities (Zhang et al., 2017; Zhang & Sabuncu, 2018; Pleiss et al., 2020; Lukasik et al., 2020; Chen et al., 2022).

To reduce the influence of noisy labels, a lot of approaches have been recently proposed (Natarajan et al., 2013; Liu & Tao, 2016; Ma et al., 2018; Yang et al., 2021b; Zheng et al., 2020; Xia et al., 2019; 2020; Tanaka et al., 2018; Malach & Shalev-Shwartz, 2017; Li et al., 2020b; Menon et al., 2018; Thekumparampil et al., 2018; Xu et al., 2019; Kim et al., 2019; Jiang et al., 2020; Harutyunyan et al., 2020). They can be generally divided into two main categories. The first one is to estimate the noise transition matrix (Patrini et al., 2017; Shu et al., 2020; Hendrycks et al., 2018; Yang et al., 2021c; Wu et al., 2022), which denotes the probabilities that clean labels flip into noisy labels. However, the noise transition matrix is hard to be estimated accurately, especially when the number of classes is large (Yu et al., 2019). The second approach is sample selection, which is *our focus* in this paper. This approach is based on selecting possibly clean examples from a mini-batch for training (Han et al., 2018; Wang et al., 2018; Yao et al., 2020a; Wang et al., 2019; Yu et al., 2019; Lee et al., 2019; Wang et al., 2019; Yao et al., 2022). Intuitively, if we can exploit less noisy data for network parameter updates, the network will be more robust.

A major question in sample selection is what *criteria* can be used to select possibly clean examples. At the present stage, the selection based on the *small-loss* criteria is the most common method, and has been verified to be effective in many circumstances (Han et al., 2018; Jiang et al., 2018; Yu et al.,

---

[†]Corresponding author

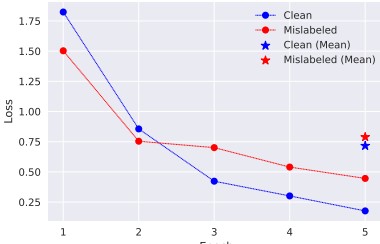 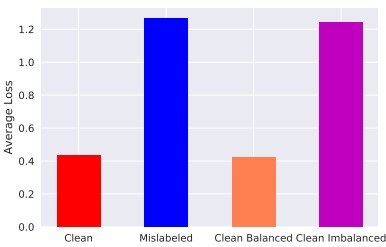

Figure 1: Illustrations of *uncertainty of losses*. Experiments are conducted on the imbalanced noisy *MNIST* dataset. **Left**: uncertainty of *small-loss* examples. At the beginning of training (Epochs 1 and 2), due to the instability of the current prediction, the network gives a larger loss to the clean example and does not select it for updates. If we consider the *mean* of training losses at different epochs, the clean example can be equipped with a smaller loss and then selected for updates. **Right**: uncertainty of *large-loss* examples. Since the deep network learns easy examples at the beginning of training, it gives a large loss to *clean imbalanced* data with non-dominant labels, which causes such data unable to be selected and severely influence generalization.

2019; Wei et al., 2020; Yao et al., 2020a). Specifically, since deep networks *learn patterns first* (Arpit et al., 2017), they would first memorize training data of clean labels and then those of noisy labels with the assumption that clean labels are of the majority in a noisy class. Small-loss examples can thus be regarded as clean examples *with high probability*. Therefore, in each iteration, prior methods (Han et al., 2018; Wei et al., 2020) select the small-loss examples based on *the predictions of the current network* for robust training.

However, such a selection procedure is *debatable*, since it arguably does *not consider uncertainty* in selection. The uncertainty comes from two aspects. First, this procedure has *uncertainty about small-loss examples*. Specifically, the procedure uses *limited time intervals* and only exploits the losses provided by the *current predictions*. For this reason, the estimation for the noisy class posterior is *unstable* (Yao et al., 2020b), which causes the network predictions to be equally unstable. It thus *takes huge risks* to only use losses provided by the current predictions (Figure 1, left). Once wrong selection is made, the inferiority of accumulated errors will arise (Yu et al., 2019). Second, this procedure has *uncertainty about large-loss examples*. To be specific, deep networks learn easy examples at the beginning of training, but ignore some clean examples with large losses. Nevertheless, such examples are always critical for generalization. For instance, when learning with *imbalanced* data, distinguishing the examples with *non-dominant labels* are more pivotal during training (Menon et al., 2020; Wei et al., 2021). Deep networks often give large losses to such examples (Figure 1, right). Therefore, when learning under the realistic scenes, e.g., learning with noisy imbalanced data, prior sample selection methods cannot address such an issue well.

To relieve the above issues, we study the uncertainty of losses in the sample selection procedure to combat noisy labels. To reduce the uncertainty of small-loss examples, we extend time intervals and utilize the *mean* of training losses at different training iterations. In consideration of the bad influence of mislabeled data on training losses, we build two *robust mean estimators* from the perspectives of *soft truncation* and *hard truncation* w.r.t. the truncation level, respectively. Soft truncation makes the mean estimation more robust by *holistically* changing the behavior of losses. Hard truncation makes the mean estimation more robust by *locally* removing outliers from losses. To reduce the uncertainty of large-loss examples, we encourage networks to pick the sample that has not been selected in a conservative way. Furthermore, to address the two issues *simultaneously*, we derive *concentration inequalities* (Boucheron et al., 2013) for robust mean estimation and further employ statistical *confidence bounds* (Auer, 2002) to consider the number of times an example was selected during training.

The study of uncertainty of losses in learning with noisy labels can be justified as follows. In statistical learning, it is known that uncertainty is related to the quality of data (Vapnik, 2013). Philosophically, we need *variety decrease* for selected data and *variety search* for unselected data, which share a common objective, i.e., *reduce the uncertainty of data to improve generalization* (Moore, 1990). This is our original intention, since noisy labels could bring more uncertainty because of the low quality of noisy data. Nevertheless, due to the harm of noisy labels for generalization, we need to strike a good balance between variety decrease and search. Technically, our method is specially designed for

handling noisy labels, which robustly uses network predictions and conservatively seeks less selected examples meanwhile to reduce the uncertainty of losses and then generalize well.

Before delving into details, we clearly emphasize our contributions in two folds. First, we reveal prior sample selection criteria in learning with noisy labels have some potential weaknesses and discuss them in detail. The new selection criteria are then proposed with detailed theoretical analyses. Second, we experimentally validate the proposed method on both synthetic noisy balanced/imbalanced datasets and real-world noisy datasets, on which it achieves superior robustness compared with the state-of-the-art methods in learning with noisy labels. The rest of the paper is organized as follows. In Section 2, we propose our robust learning paradigm step by step. Experimental results are discussed in Section 3. The conclusion is given in Section 4.

## 2 METHOD

In this section, we first introduce the problem setting and some background (Section 2.1). Then we discuss how to exploit training losses at different iterations (Section 2.2). Finally, we introduce the proposed method, which exploits training losses at different iterations more robustly and encourages networks to pick the sample that is less selected but could be correctly labeled (Section 2.3).

### 2.1 PRELIMINARIES

Let $\mathcal{X}$ and $\mathcal{Y}$ be the input and output spaces. Consider a $k$-class classification problem, i.e., $\mathcal{Y} = [k]$, where $[k] = \{1, \ldots, k\}$. In learning with noisy labels, the training data are all sampled from a corrupted distribution on $\mathcal{X} \times \mathcal{Y}$. We are given a sample with noisy labels, i.e., $\tilde{S} = \{(\mathbf{x}, \tilde{y})\}$, where $\tilde{y}$ is the noisy label. The aim is to learn a robust classifier that could assign clean labels to test data by only exploiting a training sample with noisy labels.

Let $f : \mathcal{X} \to \mathbb{R}^k$ be the classifier with learnable parameters $\mathbf{w}$. At the $i$-th iteration during training, the parameters of the classifier $f$ can be denoted as $\mathbf{w}_i$. Let $\ell : \mathbb{R}^k \times \mathcal{Y} \to \mathbb{R}$ be a *surrogate loss function* for $k$-class classification. We exploit the *softmax cross entropy loss* in this paper. Given an arbitrary training example $(\mathbf{x}, \tilde{y})$, at the $i$-th iteration, we can obtain a loss $\ell_i$, i.e., $\ell_i = \ell(f(\mathbf{w}_i; \mathbf{x}), \tilde{y})$. Hence, until the $t$-th iteration, we can obtain a training loss set $L_t$ about the example $(\mathbf{x}, \tilde{y})$, i.e., $L_t = \{\ell_1, \ldots, \ell_t\}$.

In this paper, we assume that the training losses in $L_t$ conform to a *Markov process*, which is to represent a changing system under the assumption that future states only depend on the current state (the Markov property). More specifically, at the $i$-th iteration, if we exploit an optimization algorithm for parameter updates (e.g., the stochastic gradient descent algorithm (Bottou, 2012)) and omit other dependencies (e.g., $\tilde{S}$), we will have $P(\mathbf{w}_i | \mathbf{w}_{i-1}, \ldots, \mathbf{w}_0) = P(\mathbf{w}_i | \mathbf{w}_{i-1})$, which means that the future state of the classifier $f$ only depends on the current state. Furthermore, given a training example and the parameters of the classifier $f$, we can determine the loss of the training example as discussed. Therefore, the training losses in $L_t$ will also conform to a Markov process.

### 2.2 EXTENDED TIME INTERVALS

As limited time interval cannot address the instability issue of the estimation for the noisy class posterior well (Pleiss et al., 2020), we extend time intervals and exploit the training losses at different training iterations for sample selection. One straightforward idea is to use the *mean* of training losses at different training iterations. Hence, the selection criterion could be

$$\tilde{\mu} = \frac{1}{t} \sum_{i=1}^{t} \ell_i. \tag{1}$$

It is intuitive and reasonable to use such a selection criterion for sample selection, since the operation of averaging can mitigate the risks caused by the unstable estimation for the noisy class posterior, following better generalization. Nevertheless, such a method could arguably achieve suboptimal classification performance for learning with noisy labels. The main reason is that, due to the great harm of mislabeled data, part of training losses are with too large uncertainty and could be seen as outliers. Therefore, it could be biased to use the mean of training losses consisting of such outliers (Diakonikolas et al., 2020), which further influences sample selection. More evaluations for our claims are provided in Section 3.

## 2.3 ROBUST MEAN ESTIMATION AND CONSERVATIVE SEARCH

We extend time intervals and meanwhile exploit the training losses at different training iterations more robustly. Specifically, we build two robust mean estimators from the perspectives of *soft truncation* and *hard truncation* (Catoni, 2012). Note that for specific tasks, it is feasible to decide the types of robust mean estimation with statistical tests based on some assumptions (Chakrabarty & Samorodnitsky, 2012). We leave the analysis as future work. Two *distribution-free* robust mean estimators are introduced as follows.

**Soft truncation.** We extend a classical M-estimator from (Catoni, 2012) and exploit the *widest possible choice* of the *influence function*. More specifically, give a random variable $X$, let us consider a non-decreasing influence function $\psi : \mathbb{R} \to \mathbb{R}$ such that

$$\psi(X) = \log(1 + X + X^2/2), X \geq 0. \tag{2}$$

The choice of $\psi$ is inspired by the *Taylor expansion of the exponential function*, which can make the estimation results more robust by reducing the side effect of extremum *holistically*. The illustration for this influence function is provided in Appendix A.1. For our task, given the observations on training losses, i.e., $L_t = \{\ell_1, \ldots, \ell_t\}$, we estimate the mean robustly as follows:

$$\tilde{\mu}_s = \frac{1}{t} \sum_{i=1}^{t} \psi(\ell_i). \tag{3}$$

We term the above robust mean estimator (3) the *soft estimator*.

**Hard truncation.** We propose a new robust mean estimator based on hard truncation. Specifically, given the observations on training losses $L_t$, we first exploit the K-nearest neighbor (KNN) algorithm (Liao & Vemuri, 2002) to remove some underlying outliers in $L_t$. The number of outliers is denoted by $t_o(t_o < t)$, which can be *adaptively determined* as discussed in (Zhao et al., 2019). Note that we can also employ other algorithms, e.g., principal component analysis (Shyu et al., 2003) and the local outlier factor (Breunig et al., 2000), to identify underlying outliers in $L_t$. The main reason we employ KNN is because of its relatively low computation costs (Zhao et al., 2019).

The truncated loss observations on training losses are denoted by $L_{t-t_o}$. We then utilize $L_{t-t_o}$ for the mean estimation. As the potential outliers are removed with high probability, the robustness of the estimation results will be enhanced. We denote such an estimated mean as $\tilde{\mu}_h$. We have

$$\tilde{\mu}_h = \frac{1}{t - t_o} \sum_{\ell_i \in L_{t-t_o}} \ell_i. \tag{4}$$

The corresponding estimator (4) is termed the *hard estimator*.

We derive concentration inequalities for the soft and hard estimators respectively. The search strategy for less selected examples and overall selection criterion are then provided. Note that we do not need to explicitly quantify the mean of training losses. We only need to sort the training examples based on the proposed selection criterion and then use the selected examples for robust training.

**Theorem 1** *Let $Z_n = \{z_1, \cdots, z_n\}$ be an observation set with mean $\mu_z$ and variance $\sigma^2$. By exploiting the non-decreasing influence function $\psi(z) = \log(1 + z + z^2/2)$. For any $\epsilon > 0$, we have*

$$\left| \frac{1}{n} \sum_{i=1}^{n} \psi(z_i) - \mu_z \right| \leq \frac{\sigma^2(n + \frac{\sigma^2 \log(\epsilon^{-1})}{n^2})}{n - \sigma^2}, \tag{5}$$

*with probability at least $1 - 2\epsilon$.*

Proof can be found in Appendix A.1.

**Theorem 2** *Let $Z_n = \{z_1, \ldots, z_n\}$ be a (not necessarily time homogeneous) Markov chain with mean $\mu_z$, taking values in a Polish state space $\Lambda_1 \times \ldots \times \Lambda_n$, and with a minimal mixing time $\tau_{\min}$. The truncated set with hard truncation is denoted by $Z_{n_o}$, with $n_o < n$. If $|z_i|$ is upper bounded by $Z$. For any $\epsilon_1 > 0$ and $\epsilon_2 > 0$, we have*

$$\left| \frac{1}{n - n_o} \sum_{z_i \in Z_n \setminus Z_{n_o}} -\mu_z \right| \leq \frac{1}{n - n_o} \left( 2Z \sqrt{2\tau_{\min} \log \frac{2}{\epsilon_1}} + \frac{2Z n_o}{n} \sqrt{2\tau_{\min} \log \frac{2n}{\epsilon_2}} \right), \tag{6}$$

*with probability at least $1 - \epsilon_1 - \epsilon_2$.*

Proof can be found in Appendix A.2. For our task, let the training loss be upper-bounded by $L$. The value of $L$ can be determined easily by training networks on noisy datasets and observing the loss distribution (Arazo et al., 2019).

**Conservative search and selection criteria.** In this paper, we will use the concentration inequalities (5) and (6) to present conservative search and the overall sample selection criterion. Specifically, we exploit their *lower bounds* and consider the selected number of examples during training. The selection of the examples that are less selected is encouraged.

Denote the number of times one example was selected by $n_t(n_t \leq t)$. Let $\epsilon = \frac{1}{2t}$. For the circumstance with soft truncation, the selection criterion is

$$\ell_s^\star = \tilde{\mu}_s - \frac{\sigma^2(t + \frac{\sigma^2 \log(2t)}{t^2})}{n_t - \sigma^2}. \tag{7}$$

Let $\epsilon_1 = \epsilon_2 = \frac{1}{2t}$, for the situation with hard truncation, by rewriting (6), the selection criterion is

$$\ell_h^\star = \tilde{\mu}_h - \frac{2\sqrt{2\tau_{\min}}L(t + \sqrt{2}t_\text{o})}{(t - t_\text{o})\sqrt{t}}\sqrt{\frac{\log(4t)}{n_t}}. \tag{8}$$

Note that we directly replace $t$ with $n_t$. If an example is rarely selected during training, $n_t$ will be far less than $n$, which causes the lower bounds to change drastically. Hence, we do not use the mean of all training losses, but use the mean of training losses in fixed-length time intervals. More details about this can be checked in Section 3.

For the selection criteria (7) and (8), we can see that they consist of two terms and have one term with a minus sign. The first term in Eq. (7) (or Eq. (8)) is to reduce the uncertainty of small-loss examples, where we use robust mean estimation on training losses. The second term, i.e., the statistical confidence bound, is to encourage the network to choose the less selected examples (with a small $n_t$). The two terms are constraining and balanced with $\sigma^2$ or $\tau_{\min}$. To avoid introducing strong assumptions on the underlying distribution of losses (Chakrabarty & Samorodnitsky, 2012), we tune $\sigma$ and $\tau_{\min}$ with a noisy validation set. For the mislabeled data, although the model has high uncertainties on them (i.e., a small $n_t$) and tends to pick them, the overfitting to the mislabeled data is harmful. Also, the mislabeled data and clean data are rather hard to distinguish in some cases as discussed. Thus, we should search underlying clean data in a conservative way. In this paper, we initialize $\sigma$ and $\tau_{\min}$ with small values. This way can reduce the adverse effects of mislabeled data and meanwhile select the clean examples with large losses, which helps generalize. More evaluations will be presented in Section 3.

The overall procedure of the proposed method, which **c**ombats **n**oisy **l**abels by **c**oncerning **u**ncertainty (CNLCU), is provided in Algorithm 1. CNLCU works in a mini-batch manner since all deep learning training methods are based on stochastic gradient descent. Following (Han et al., 2018), we exploit two networks with parameters $\theta_1$ and $\theta_2$ respectively to teach each other. Specifically, when a mini-batch $\bar{S}$ is formed (Step 3), we let two networks select a small proportion of examples in this mini-batch with Eq. (7) or (8) (Step 4 and Step 5). The number of instances is controlled by the function $R(T)$, and two networks only select $R(T)$ percentage of examples out of the mini-batch. The value of $R(T)$ should be larger at the beginning of

---

**Algorithm 1** CNLCU Algorithm.

1: **Input** $\theta_1$ and $\theta_2$, learning rate $\eta$, fixed $\tau$, epoch $T_k$ and $T_{\max}$, iteration $t_{\max}$;
**for** $T = 1, 2, \ldots, T_{\max}$ **do**
    2: **Shuffle** training dataset $\tilde{S}$;
    **for** $t = 1, \ldots, t_{\max}$ **do**
        3: **Fetch** mini-batch $\bar{S}$ from $\tilde{S}$;
        4: **Obtain** $\bar{S}_1 = \arg\min_{S':|S'|\geq R(T)|\bar{S}|} \ell^\star(\theta_1, S')$;
        // calculated with Eq. (7) or Eq. (8)
        5: **Obtain** $\bar{S}_2 = \arg\min_{S':|S'|\geq R(T)|\bar{S}|} \ell^\star(\theta_2, S')$;
        // calculated with Eq. (7) or Eq. (8)
        6: **Update** $\theta_1 = \theta_1 - \eta\nabla\ell(\theta_1, \bar{S}_2)$;
        7: **Update** $\theta_2 = \theta_2 - \eta\nabla\ell(\theta_2, \bar{S}_1)$;
    **end**
    8: **Update** $R(T) = 1 - \min\left\{\frac{T}{T_k}\tau, \tau\right\}$;
**end**
9: **Output** $\theta_1$ and $\theta_2$.

---

training, and be smaller when the number of epochs goes large, which can make better use of memorization effects of deep networks (Han et al., 2018) for sample selection. Then, the selected instances are fed into its peer network for parameter updates (Step 6 and Step 7).

## 3 EXPERIMENTS

In this section, we evaluate the robustness of our proposed method to noisy labels with comprehensive experiments on the synthetic balanced noisy datasets (Section 3.1), synthetic imbalanced noisy datasets (Section 3.2), and real-world noisy dataset (Section 3.3).

### 3.1 EXPERIMENTS ON SYNTHETIC BALANCED NOISY DATASETS

**Datasets.** We verify the effectiveness of our method on the manually corrupted version of the following datasets: *MNIST* (LeCun et al.), *F-MNIST* (Xiao et al., 2017), *CIFAR-10* (Krizhevsky, 2009), and *CIFAR-100* (Krizhevsky, 2009), because these datasets are popularly used for the evaluation of learning with noisy labels in the literature (Han et al., 2018; Yu et al., 2019; Wu et al., 2021; Lee et al., 2019). The four datasets are *class-balanced*. The important statistics of the used synthetic datasets are summarized in Appendix B.1.

**Generating noisy labels.** We consider broad types of label noise: (1). Symmetric noise (abbreviated as Sym.) (Wu et al., 2020; Ma et al., 2018). (2) Asymmetric noise (abbreviated as Asym.) (Ma et al., 2020; Xia et al., 2021; Wei et al., 2020). (3) Pairflip noise (abbreviated as Pair.) (Han et al., 2018; Yu et al., 2019; Zheng et al., 2020). (4). Tridiagonal noise (abbreviated as Trid.) (Zhang et al., 2021). (5). Instance noise (abbreviated as Ins.) (Cheng et al., 2020; Xia et al., 2020). The noise rate is set to 20% and 40% to ensure clean labels are diagonally dominant (Ma et al., 2020). More details about above noise are provided in Appendix B.1. We leave out 10% of noisy training examples as a validation set.

**Baselines.** We compare the proposed method (Algorithm 1) with following methods which focus on sample selection, and implement all methods with default parameters by PyTorch, and conduct all the experiments on NVIDIA Titan Xp GPUs. (1). S2E (Yao et al., 2020a), which properly controls the sample selection process so that deep networks can better benefit from the memorization effects. (2). MentorNet (Jiang et al., 2018), which learns a curriculum to filter out noisy data. We use self-paced MentorNet in this paper. (3). Co-teaching (Han et al., 2018), which trains two networks simultaneously and cross-updates parameters of peer networks. (4). SIGUA (Han et al., 2020), which exploits stochastic integrated gradient underweighted ascent to handle noisy labels. We use self-teaching SIGUA in this paper. (5). JoCor (Wei et al., 2020), which reduces the diversity of networks to improve robustness. To avoid too dense tables, we provide results of other sample selection methods and other types of baselines such as *adding regularization*. All results are presented in Appendix B.2. Here, we term our methods with soft truncation and hard truncation as CNLCU-S and CNLCU-H respectively.

**Network structure and optimizer.** For *MNIST*, *F-MNIST*, and *CIFAR-10*, we use a 9-layer CNN structure from (Han et al., 2018). Due to the limited space, the experimental details on *CIFAR-100* are provided in Appendix B.3. All network structures we used here are standard test beds for weakly-supervised learning. For all experiments, the Adam optimizer (Kingma & Ba, 2014) (momentum=0.9) is used with an initial learning rate of 0.001, and the batch size is set to 128 and we run 200 epochs. We linearly decay learning rate to zero from 80 to 200 epochs as did in (Han et al., 2018). We take two networks with the same architecture but different initializations as two classifiers as did in (Han et al., 2018; Yu et al., 2019; Wei et al., 2020), since even with the same network and optimization method, different initializations can lead to different local optimal (Han et al., 2018). The details of network structures can be checked in Appendix C.

For the hyper-parameters $\sigma^2$ and $\tau_{\min}$, we determine them in the range $\{10^{-1}, 10^{-2}, 10^{-3}, 10^{-4}\}$ with a noisy validation set. Note that the use of hyperparameters aims to reduce the dependency on strong assumptions and thus make our methods perform well in practice. We provide more details about this in Appendix D. Here, we assume the noise level $\tau$ is known and set $R(T) = 1 - \min\{\frac{T}{T_k}\tau, \tau\}$ with $T_k$=10. If $\tau$ is not known in advanced, it can be inferred using validation sets (Liu & Tao, 2016; Yu et al., 2018). As for performance measurement, we use test accuracy, i.e., *test accuracy = (# of correct prediction) / (# of testing)*. All experiments are repeated five times. We report the mean and standard deviation of experimental results.

**Experimental results.** The experimental results about test accuracy are provided in Tables 1, 2, and 3. Specifically, for *MNIST*, as can be seen, our proposed methods, i.e., CNLCU-S and CNLCU-H, produce the best results in the vast majority of cases. In some cases such as asymmetric noise, the baseline S2E outperforms ours, which benefits the accurate estimation for the number of selected

| Noise type | Sym. | | Asym. | | Pair. | | Trid. | | Ins. | |
|---|---|---|---|---|---|---|---|---|---|---|
| Method/Noise | 20% | 40% | 20% | 40% | 20% | 40% | 20% | 40% | 20% | 40% |
| S2E | 98.46 | 95.62 | **99.05** | **98.45** | **98.56** | 94.22 | **99.02** | 97.23 | 97.93 | 94.02 |
| | ±0.06 | ±0.91 | ±**0.02** | ±**0.26** | ±**0.32** | ±0.79 | ±**0.09** | ±1.26 | ±1.26 | ±2.39 |
| MentorNet | 95.04 | 92.08 | 96.32 | 90.86 | 93.19 | 90.93 | 96.42 | 93.28 | 94.65 | 90.11 |
| | ±0.03 | ±0.42 | ±0.17 | ±0.97 | ±0.17 | ±1.54 | ±0.09 | ±1.37 | ±0.73 | ±1.26 |
| Co-teaching | 97.53 | 95.62 | 98.25 | 95.08 | 96.05 | 94.16 | 98.05 | 96.18 | 97.96 | 95.02 |
| | ±0.12 | ±0.30 | ±0.08 | ±0.43 | ±0.96 | ±1.37 | ±0.06 | ±0.85 | ±0.09 | ±0.39 |
| SIGUA | 92.31 | 91.88 | 93.96 | 62.59 | 93.77 | 86.22 | 94.92 | 83.46 | 92.90 | 86.34 |
| | ±1.10 | ±0.92 | ±0.82 | ±0.15 | ±1.40 | ±1.75 | ±0.83 | ±2.98 | ±1.82 | ±3.51 |
| JoCor | 98.42 | 98.04 | 98.05 | 94.55 | 98.01 | 96.85 | 98.45 | 96.98 | 98.62 | 96.07 |
| | ±0.14 | ±0.07 | ±0.37 | ±1.08 | ±0.19 | ±0.43 | ±0.17 | ±0.25 | ±0.06 | ±0.31 |
| CNLCU-S | **98.82** | **98.31** | 98.93 | 97.67 | **98.86** | **97.71** | **99.09** | **98.02** | **98.77** | **97.78** |
| | ±**0.03** | ±**0.05** | ±0.06 | ±0.22 | ±**0.06** | ±**0.64** | ±**0.04** | ±**0.17** | ±**0.08** | ±**0.25** |
| CNLCU-H | **98.70** | **98.24** | **99.01** | **98.01** | 98.44 | **97.37** | 98.89 | **97.92** | **98.74** | **97.42** |
| | ±**0.06** | ±**0.06** | ±**0.04** | ±**0.03** | ±0.19 | ±**0.32** | ±0.15 | ±**0.05** | ±**0.16** | ±**0.39** |

Table 1: Test accuracy (%) on *MNIST* over the last ten epochs. The best two results are in bold.

| Noise type | Sym. | | Asym. | | Pair. | | Trid. | | Ins. | |
|---|---|---|---|---|---|---|---|---|---|---|
| Method/Noise | 20% | 40% | 20% | 40% | 20% | 40% | 20% | 40% | 20% | 40% |
| S2E | 89.99 | 75.32 | 89.00 | 81.03 | 88.66 | 67.09 | 89.53 | 77.29 | 88.65 | 79.35 |
| | ±2.07 | ±5.84 | ±0.95 | ±1.93 | ±1.32 | ±4.03 | ±2.63 | ±3.97 | ±2.12 | ±3.04 |
| MentorNet | 90.37 | 86.53 | 89.69 | 67.21 | 87.92 | 83.70 | 88.74 | 85.63 | 87.52 | 83.27 |
| | ±0.17 | ±0.65 | ±0.19 | ±2.94 | ±1.08 | ±0.49 | ±0.33 | ±0.59 | ±0.15 | ±1.42 |
| Co-teaching | 91.48 | 88.80 | 91.03 | 68.07 | 90.77 | 86.91 | 91.24 | 89.18 | 90.60 | 87.90 |
| | ±0.10 | ±0.29 | ±0.14 | ±4.58 | ±0.23 | ±0.71 | ±0.11 | ±0.36 | ±0.12 | ±0.45 |
| SIGUA | 87.64 | 87.23 | 76.97 | 45.96 | 69.59 | 68.93 | 79.97 | 76.14 | 76.92 | 74.89 |
| | ±1.29 | ±0.72 | ±2.59 | ±3.40 | ±5.75 | ±2.80 | ±3.23 | ±4.24 | ±5.09 | ±4.84 |
| JoCor | 91.97 | 89.96 | 90.95 | 79.79 | 91.52 | 87.40 | 92.01 | 89.42 | 91.43 | 87.59 |
| | ±0.13 | ±0.19 | ±0.21 | ±2.39 | ±0.24 | ±0.58 | ±0.17 | ±0.33 | ±0.71 | ±0.94 |
| CNLCU-S | **92.37** | **91.45** | **92.57** | **83.14** | **92.04** | **88.20** | **92.24** | **90.08** | **91.69** | **89.02** |
| | ±**0.15** | ±**0.28** | ±**0.15** | ±**1.77** | ±**0.26** | ±**0.44** | ±**0.17** | ±**0.34** | ±**0.10** | ±**1.02** |
| CNLCU-H | **92.42** | **91.60** | **92.60** | **82.69** | **91.70** | **87.70** | **92.33** | **90.22** | **91.50** | **88.79** |
| | ±**0.21** | ±**0.19** | ±**0.18** | ±**0.43** | ±**0.18** | ±**0.69** | ±**0.26** | ±**0.71** | ±**0.21** | ±**1.22** |

Table 2: Test accuracy on *F-MNIST* over the last ten epochs. The best two results are in bold.

small-loss examples. For *F-MNIST*, the training data becomes complicated. S2E cannot achieve the accurate estimation in such situation and thus has no great performance like it got on *MNIST*. Our methods achieve varying degrees of lead over baselines. For *CIFAR-10*, our methods once again outperforms all the baseline methods. Although some baseline, e.g., Co-teaching, can work well in some cases, experimental results show that it cannot handle various noise types. In contrast, the proposed methods achieve superior robustness against broad noise types. The results mean that our methods can be better applied to actual scenarios, where the noise is diversiform.

**Ablation study.** We first conduct the ablation study to analyze the sensitivity of the length of time intervals. In order to *avoid too dense figures*, we exploit *MNIST* and *F-MNIST* with the mentioned noise settings as representative examples. For CNLCU-S, the length of time intervals is chosen in the range from 3 to 8. For CNLCU-H, the length of time intervals is chosen in the range from 10 to 15. Note that the reason for their different lengths is that their different mechanisms. Specifically, CNLCU-S holistically changes the behavior of losses, but does not remove any loss from the loss set. We thus do not need too long length of time intervals. As a comparison, CNLCU-H needs to remove some outliers from the loss set as discussed. The length should be longer to guarantee the number of examples available for robust mean estimation. The experimental results are provided in Appendix B.4, which show the proposed CNLCU-S and CNLCU-H are robust to the choices of the length of time intervals. Such robustness to hyperparameters means our methods can be applied in practice and does not need too much effect to tune the hyperparameters.

Furthermore, since our methods concern uncertainty from two aspects, i.e., the uncertainty from both small-loss and large-loss examples, we conduct experiments to analyze each part of our methods. Also, as mentioned, we compare robust mean estimation with non-robust mean estimation when learning with noisy labels. More details are provided in Appendix B.4.

| Noise type | Sym. | | Asym. | | Pair. | | Trid. | | Ins. | |
|---|---|---|---|---|---|---|---|---|---|---|
| Method/Noise | 20% | 40% | 20% | 40% | 20% | 40% | 20% | 40% | 20% | 40% |
| S2E | 80.78 | 69.72 | 84.03 | 75.04 | 81.72 | 61.50 | 81.44 | 64.39 | 79.89 | 62.42 |
| | ±0.88 | ±3.94 | ±1.01 | ±1.24 | ±0.93 | ±4.63 | ±0.59 | ±2.82 | ±0.26 | ±3.11 |
| MentorNet | 80.92 | 74.67 | 80.37 | 71.69 | 77.98 | 69.39 | 78.02 | 71.56 | 77.02 | 68.17 |
| | ±0.48 | ±1.17 | ±0.26 | ±1.06 | ±0.31 | ±1.73 | ±0.29 | ±0.93 | ±0.71 | ±2.52 |
| Co-teaching | 82.35 | 77.96 | 83.87 | 73.43 | 80.94 | 72.81 | 81.17 | **74.37** | 79.92 | 73.29 |
| | ±0.16 | ±0.39 | ±0.24 | ±0.62 | ±0.46 | ±0.92 | ±0.60 | **±0.64** | ±0.57 | ±1.62 |
| SIGUA | 78.19 | 77.67 | 75.14 | 52.76 | 74.41 | 61.91 | 75.75 | 74.05 | 74.34 | 67.98 |
| | ±0.22 | ±0.41 | ±0.36 | ±0.68 | ±0.81 | ±5.27 | ±0.53 | ±0.41 | ±0.39 | ±1.34 |
| JoCor | 80.96 | 76.65 | 81.39 | 69.92 | 80.33 | 71.62 | 79.03 | 74.33 | 78.21 | 71.46 |
| | ±0.25 | ±0.43 | ±0.74 | ±1.63 | ±0.20 | ±1.05 | ±0.13 | ±1.09 | ±0.34 | ±1.27 |
| CNLCU-S | **83.03** | **78.25** | **85.06** | **75.34** | **83.16** | **73.19** | **82.77** | 74.37 | **82.03** | **73.67** |
| | **±0.21** | **±0.70** | **±0.17** | **±0.32** | **±0.25** | **±1.25** | **±0.32** | ±1.37 | **±0.37** | **±1.09** |
| CNLCU-H | **83.03** | **78.33** | **84.95** | **75.29** | **83.39** | **73.40** | **82.52** | **74.79** | **81.93** | **73.58** |
| | **±0.47** | **±0.50** | **±0.27** | **±0.80** | **±0.68** | **±1.53** | **±0.71** | **±1.13** | **±0.25** | **±1.39** |

Table 3: Test accuracy (%) on *CIFAR-10* over the last ten epochs. The best two results are in bold.

## 3.2 EXPERIMENTS ON SYNTHETIC IMBALANCED NOISY DATASETS

**Experimental setup.** We exploit *MNIST* and *F-MNIST*. For these two datasets, we reduce the number of training examples along with the labels from "0" to "4" to 1% of previous numbers. We term such synthetic imbalanced noisy datasets as *IM-MNIST* and *IM-F-MNIST* respectively. This setting aims to simulate the extremely imbalanced circumstance, which is common in practice. Moreover, we exploit asymmetric noise, since these types of noise can produce more imbalanced case (Patrini et al., 2017; Ma et al., 2020). Other settings such as the network structure and optimizer are the same as those in experiments on synthetic balanced noisy datasets.

As for performance measurements, we use test accuracy. In addition, we exploit the selected ratio of training examples with the imbalanced classes, i.e., *selected ratio=(# of selected imbalanced labels / # of all selected labels)*. Intuitively, a higher selected ratio means the proposed method can make better use of training examples with the imbalanced classes, following better generalization (Kang et al., 2020).

**Experimental results.** The test accuracy achieved on *IM-MNIST* and *IM-F-MNIST* is presented in Figure 2. Recall the experimental results in Tables 1 and 2, we can see that the imbalanced issue is *catastrophic* to the sample selection approach when learning with noisy labels. For *IM-MNIST*, as can be seen, all the baselines have serious overfitting in the early stages of training. The curves of test accuracy drop dramatically. As a comparison, the proposed CNLCU-S and CNLCU-H can give a try to large-loss but less selected data which are possible to be clean but equipped with imbalanced labels. Therefore, our methods always outperform baselines clearly. In the case of Asym. 10%, our methods achieve nearly 30% lead over baselines. For *IM-F-MNIST*, we can also see that our methods perform well and always achieve about 5% lead over all the baselines. Note that due to the huge challenge of this task, some baseline, e.g., S2E, has a large error bar. In addition, the baseline SIGUA performs badly. It is because SIGUA exploits stochastic integrated gradient underweighted ascent on large-loss examples, which makes the examples with imbalanced classes more difficult to be selected than them in other sample selection methods. Due to the limited space, the selected ratio achieved on *IM-MNIST* and *IM-F-MNIST* is presented in Appendix B.5, which explain well why our methods perform better than multiple baselines.

## 3.3 EXPERIMENTS ON REAL-WORLD NOISY DATASETS

**Experimental setup.** To verify the efficacy of our methods in the real-world scenario, we conduct experiments on the noisy dataset *Clothing1M* (Xiao et al., 2015). Specifically, for experiments on *Clothing1M*, we use the 1M images with noisy labels for training and 10k clean data for test respectively. Note that we do not use the 50k clean training data in all the experiments. For preprocessing, we resize the image to 256×256, crop the middle 224×224 as input, and perform normalization. The experiments on *Clothing1M* are performed once due to the huge computational cost. We leave 10% noisy training data as a validation set for model selection. Note that we do not exploit the resampling trick during training (Li et al., 2020a). Here, *Best* denotes the test accuracy of the epoch where the validation accuracy was optimal. *Last* denotes test accuracy of the last epoch. For

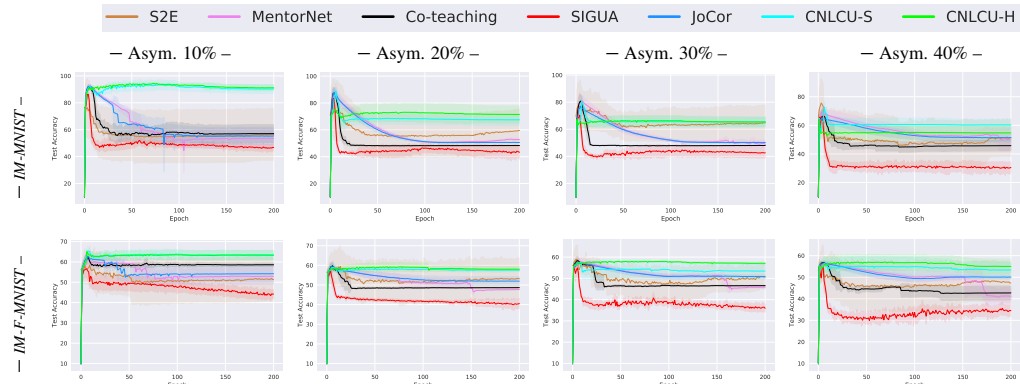

Figure 2: Test accuracy vs. number of epochs on *IM-MNIST* and *IM-F-MNIST*. The error bar for standard deviation in each figure has been shaded.

the experiments on *Clothing1M*, we use ResNet-18 and ResNet-50 which are pretrained on ImageNet. We also use the Adam optimizer and set the batch size to 64. During the training stage, we run 15 epochs in total and set the learning rate $8 \times 10^{-4}$, $5 \times 10^{-4}$, and $5 \times 10^{-5}$ for 5 epochs each.

**Experimental results.** The results on *Clothing1M* are provided in Table 4. Specifically, the proposed methods get better results than state-of-the-art methods on *Best*. With ResNet-18, we achieve improvements of +1.28% and +0.99%. With ResNet-50, we achieve improvements of +2.51% and +2.16%. Likewise, the proposed methods outperform all the baselines on *Last*. We achieve improvements of +1.01% and +0.54% with ResNet-18, and improvements of +2.47% and +2.05% with ResNet-50. All these results verify the effectiveness of the proposed methods.

| Methods | S2E | MentorNet | Co-teaching | SIGUA | JoCor | CNLCU-S | CNLCU-H |
|---|---|---|---|---|---|---|---|
| *Best* (R-18) | 67.34 | 68.36 | 69.37 | 62.89 | 70.09 | **71.37** | **71.08** |
| *Last* (R-18) | 65.90 | 67.42 | 68.62 | 58.73 | 69.75 | **70.76** | **70.29** |
| *Best* (R-50) | 68.03 | 67.25 | 67.94 | 65.37 | 69.06 | **71.57** | **71.22** |
| *Last* (R-50) | 66.25 | 66.59 | 67.05 | 60.77 | 68.41 | **70.88** | **70.46** |

Table 4: Test accuracy (%) on *Clothing1M*. "R-18" (resp. "R-50") means that we exploit ResNet-18 (resp. ResNet-50). The best two results are in bold.

**Combining with semi-supervised learning.** For combating noisy labels in real-world noisy datasets, the state-of-the-art methods, e.g., DivideMix (Li et al., 2020a), always employ the semi-supervised learning technology. As our methods mainly focus on sample selection, to make the comparison fair, we combine our methods with

| Methods | *Food-101* | *WebVision (Mini)* | *Clothing1M* |
|---|---|---|---|
| DivideMix | 86.73 | 77.32 | 74.76 |
| DivideMix-S | **86.92** | **77.53** | **74.90** |
| DivideMix-H | **86.88** | **77.48** | **74.82** |

Table 5: The test accuracy (%) on three real-world datasets. DivideMix-S (resp. DivideMix-H) means that our CNLCU-S (resp. CNLCU-H) is combined with the advanced techniques in DivideMix. The best two results are in bold.

semi-supervised learning. The sample selection procedure in DivideMix is replaced by our methods. Other settings are kept the same. Following prior works (Ma et al., 2020), the experiments are conducted on three real-world noisy datasets, i.e., *Food-101* (Bossard et al., 2014), *WebVision (Mini)* (Li et al., 2017), and *Clothing1M* (Xiao et al., 2015). The results are provided in Table 5. As can be seen, the proposed methods are superior and can be used to improve the cutting edge performance.

## 4 CONCLUSION

In this paper, we focus on promoting the prior sample selection in learning with noisy labels, which starts from concerning the uncertainty of losses during training. We robustly use the training losses at different iterations to reduce the uncertainty of small-loss examples, and adopt confidence interval estimation to reduce the uncertainty of large-loss examples. Experiments are conducted on benchmark datasets, demonstrating the effectiveness of our method. We believe that this paper opens up new possibilities in the topics of using sample selection to handle noisy labels, especially in improving the robustness of models on imbalanced noisy datasets.

## 5 ETHICS STATEMENT

This paper doesn't raise any ethics concerns. This study doesn't involve any human subjects, practices to data set releases, potentially harmful insights, methodologies and applications, potential conflicts of interest and sponsorship, discrimination/bias/fairness concerns, privacy and security issues, legal compliance, and research integrity issues.

## ACKNOWLEDGMENT

XBX is supported by Australian Research Council Projects DE-190101473. TLL is partially supported by Australian Research Council Projects DE-190101473, IC-190100031, and DP-220102121. BH was supported by the RGC Early Career Scheme No. 22200720 and NSFC Young Scientists Fund No. 62006202. MMG is supported by Australian Research Council Projects DE-210101624. JY is sponsored by CAAI-Huawei MindSpore Open Fund (CAAIXSJLJJ-2021-016B). GN and MS are supported by JST AIP Acceleration Research Grant Number JPMJCR20U3, Japan. MS is also supported by the Institute for AI and Beyond, UTokyo.

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

## APPENDIX

The appendices are organized as follows. Section A details the proofs for our theorems. Section B presents more experimental results. Section C shows the details of used network structures. Section D gives more justifications for our theoretical analyses. Section E provides more discussions on learning by considering uncertainty. Section F provides more discussions on sample selection methods.

## A   PROOF OF THEORETICAL RESULTS

### A.1   PROOF OF THEOREM 1

For the circumstance with soft truncation, $\tilde{\mu}_z = \frac{1}{n} \sum_{i=1}^{n} \psi(z_i)$. As suggested in (Catoni, 2012), we can exploit $\tilde{\mu}_z^-$ and $\tilde{\mu}_z^+$ such that

$$\tilde{\mu}_z^- \leq \tilde{\mu}_z \leq \tilde{\mu}_z^+, \tag{9}$$

to derive a bound for $\tilde{\mu}_z$. For some positive real parameter $\alpha$, we define

$$r(\tilde{\mu}_z) = \sum_{i=1}^{n} \psi\left[\alpha(z_i - \tilde{\mu}_z)\right] = 0. \tag{10}$$

Let us introduce the quantity

$$r(\theta) = \frac{1}{\alpha n} \sum_{i=1}^{n} \psi\left[\alpha(z_i - \theta)\right]. \tag{11}$$

With the exponential moment inequality (Giné et al., 2000) and the $C_r$ inequality (Mohri et al., 2018), we have

$$
\begin{aligned}
\exp\{\alpha n r(\theta)\} &\leq \left\{1 + \alpha(\mu_z - \theta) + \alpha^2[\sigma^2 + (\mu_z - \theta)^2]\right\}^n \\
&\leq \exp\{n\alpha(\mu_z - \theta) + n\alpha^2[\sigma^2 + (\mu_z - \theta)^2]\}.
\end{aligned}
\tag{12}
$$

In the same way,

$$\exp\{-\alpha n r(\theta)\} \leq \exp\{-n\alpha(\mu_z - \theta) + n\alpha^2[\sigma^2 + (\mu_z - \theta)^2]\}. \tag{13}$$

If we define for any $\mu_s \in \mathbb{R}$ the bounds

$$B_-(\theta) = \mu_z - \theta - \alpha[\sigma^2 + (\mu_z - \theta)^2] - \frac{\log(\epsilon^{-1})}{\alpha n} \tag{14}$$

and

$$B_+(\theta) = \mu_z - \theta + \alpha[\sigma^2 + (\mu_z - \theta)^2] + \frac{\log(\epsilon^{-1})}{\alpha n}. \tag{15}$$

From (Chen et al., 2020) (Lemma 2.2), we obtain that

$$P(r(\theta) > B_-(\theta)) \geq 1 - \epsilon \quad \text{and} \quad P(r(\theta) < B_+(\theta)) \geq 1 - \epsilon. \tag{16}$$

Let $\tilde{\mu}_z^-$ be the largest solution of the quadratic equation $B_-(\theta)$ and $\tilde{\mu}_z^+$ be the smallest solution of the quadratic equation $B_+(\theta)$. Also, to guarantee the solution of the quadratic equation, we assume

$$4\alpha^2\sigma^2 + \frac{4\log(\epsilon^{-1})}{n} \leq 1. \tag{17}$$

From (Chen et al., 2020) (Theorem 2.6), we then have

$$\tilde{\mu}_z^- \geq \mu_z - \frac{\alpha\sigma^2 + \frac{\log(\epsilon^{-1})}{\alpha n}}{\alpha - 1}, \tag{18}$$

and

$$\tilde{\mu}_z^+ \leq \mu_z + \frac{\alpha\sigma^2 + \frac{\log(\epsilon^{-1})}{\alpha n}}{\alpha - 1}. \tag{19}$$

With probability at least 1-2$\epsilon$, we have $\tilde{\mu}_z^- \leq \tilde{\mu}_z \leq \tilde{\mu}_z^+$. We can choose $\alpha = \frac{n}{\sigma^2}$. Then we have

$$|\tilde{\mu}_z - \mu_z| \leq \frac{\sigma^2(n + \frac{\sigma^2 \log(\epsilon^{-1})}{n^2})}{n - \sigma^2}, \tag{20}$$

which holds with probability at least 1-2$\epsilon$.

We exploit the lower bound and let $\epsilon = \frac{1}{2t}$. Then we have

$$\ell_s^\star = \tilde{\mu}_s - \frac{\sigma^2(t + \frac{\sigma^2 \log(2t)}{t^2})}{n_t - \sigma^2}, \tag{21}$$

where $n_t$ denotes the number of times that the example was selected in the time intervals.

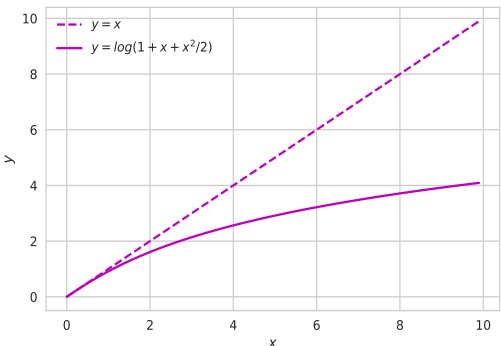

Figure 3: The illustration of the influence function for the soft estimator.

Here, we provide the graph of the used influence function for the soft estimator, which explains the mechanism of the function $y = \log(1 + x + x^2/2)$ more clearly. The illustration is presented in Figure 3. As can be seen, when $x$ is large and may be an outlier, the influence function can reduce its negative impact for mean estimation. Therefore, we exploit such an influence function for robust mean estimation, which brings better classification performance.

## A.2 PROOF OF THEOREM 2

**Lemma 1 ((Paulin et al., 2015))** *Let $Z_n = \{z_1, \ldots, z_n\}$ be a (not necessarily time homogeneous) Markov chain with mean $\mu_z$, taking values in a Polish state space $\Lambda_1 \times \ldots \times \Lambda_n$, with a mixing time $\tau(\upsilon)$ (for $0 \leq \upsilon \leq 1$). Let*

$$\tau_{\min} = \inf_{0 \leq \upsilon < 1} \tau(\upsilon) \cdot \left(\frac{2 - \upsilon}{1 - \upsilon}\right)^2. \tag{22}$$

*For some $\eta \in \mathbb{R}^+$, suppose that $f : \Lambda \to \mathbb{R}$ satisfies the following inequality:*

$$f(a) - f(b) \leq \sum_{i=1}^{n} \eta \mathbb{1}[a_i \neq b_i], \tag{23}$$

*for every $a, b \in \Lambda$. Then for any $\epsilon \geq 0$, we have*

$$P(|f(Z_n) - \mathbb{E}f(Z_n)| \geq \epsilon) \leq 2 \exp\left(\frac{-2\epsilon^2}{\eta^2 \tau_{\min}}\right). \tag{24}$$

The detailed definition of the mixing time for the Markov chain can be found in (Paulin et al., 2015; Roberts et al., 2004). Let $f$ be the mean function. Following the prior work on mean estimation (Liu et al., 2019; Diakonikolas et al., 2020; Diakonikolas & Kane, 2019; Niss & Tewari, 2020), without loss of generality, we assume $\mu_z = 0$ for the underlying true distribution, and $|z_i|$ is upper bounded by $Z$. Then we can set $\eta$ to $4Z/n$ for Eq. (23). Combining the above analyses, we can revise Eq. (24) as follows:

$$P\left(\left|\frac{1}{n}\sum_{i=1}^{n} z_i\right| \geq \frac{2Z}{n}\sqrt{2\tau_{\min}\log\frac{2}{\epsilon_1}}\right) \leq \epsilon_1, \tag{25}$$

and

$$P\left(\max_{i\in[n]}|z_i| \geq \frac{2Z}{n}\sqrt{2\tau_{\min}\log\frac{2n}{\epsilon_2}}\right) \leq \epsilon_2, \tag{26}$$

for $\epsilon_1 > 0$ and $\epsilon_2 > 0$. If we remove the potential outliers $Z_{n_o}$ from $Z_n$. Therefore, we have

$$\begin{aligned}
\left|\frac{1}{n-n_o}\sum_{z_i\in Z_n\setminus Z_{n_o}} -\mu_z\right| &= \frac{1}{n-n_o}\left|\sum_{z_i\in Z_n} - \sum_{z_i\in Z_{n_o}}\right| \\
&\leq \frac{1}{n-n_o}\left(\left|\sum_{z_i\in Z_n}\right| + \left|\sum_{z_i\in Z_{n_o}}\right|\right) \\
&\leq \frac{1}{n-n_o}\left(\left|\sum_{z_i\in Z_n}\right| + n_o\max_{i\in[n]}|z_i|\right) \\
&\leq \frac{1}{n-n_o}\left(2Z\sqrt{2\tau_{\min}\log\frac{2}{\epsilon_1}} + \frac{2Zn_o}{n}\sqrt{2\tau_{\min}\log\frac{2n}{\epsilon_2}}\right),
\end{aligned} \tag{27}$$

which holds with probability at least $1 - \epsilon_1 - \epsilon_2$.

For our task, we exploit the concentration inequality. Let $\epsilon_1 = \epsilon_2 = \frac{1}{2t}$, and the losses be bounded by $L$. Next we can obtain

$$\begin{aligned}
|\tilde{\mu}_h - \mu| &\leq \frac{2L}{t-t_o}\left(\sqrt{2\tau_{\min}\log(4t)} + \frac{t_o}{t}\sqrt{4\tau_{\min}\log(4t)}\right) \\
&= \frac{2\sqrt{2\tau_{\min}}L(t+\sqrt{2}t_o)}{(t-t_o)t}\sqrt{\log(4t)}
\end{aligned} \tag{28}$$

with the probability at least $1 - \frac{1}{t}$. In practice, it is easy to identify the value of $L$. For example, we can training deep networks on noisy datasets to observe the loss distributions. Then, we exploit the lower bound such that

$$\ell_h^\star = \tilde{\mu}_h - \frac{2\sqrt{2\tau_{\min}}L(t+\sqrt{2}t_o)}{(t-t_o)\sqrt{t}}\sqrt{\frac{\log(4t)}{n_t}} \tag{29}$$

for sample selection.

## B  COMPLEMENTARY EXPERIMENTAL ANALYSES

| | # of training | # of testing | # of class | size |
|---|---|---|---|---|
| *MNIST* | 60,000 | 10,000 | 10 | 28×28×1 |
| *F-MNIST* | 60,000 | 10,000 | 10 | 28×28×1 |
| *CIFAR-10* | 50,000 | 10,000 | 10 | 32×32×3 |
| *CIFAR-100* | 50,000 | 10,000 | 100 | 32×32×3 |

Table 6: Summary of synthetic datasets used in the experiments.

### B.1  THE DETAILS OF DATASETS AND GENERATING NOISY LABELS

For the details of datasets, the important statistics of the used datasets are summarized in Table 6.

For the details of generating noisy labels, we exploit both *class-dependent* and *instance-dependent label noise* which include five types of synthetic label noise to verify the effectiveness of the proposed method. Here, we describe the details of the noise setting as follows:

(1). Class-dependent label noise:

• Symmetric noise: this kind of label noise is generated by flipping labels in each class uniformly to incorrect labels of other classes.

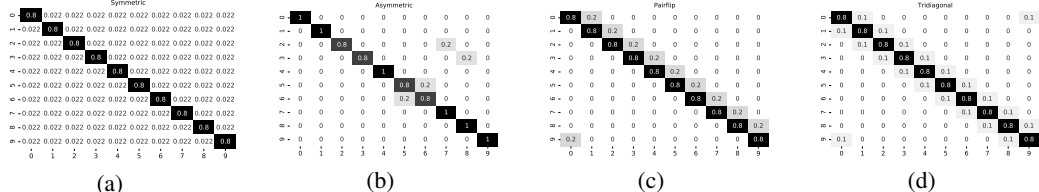

Figure 4: Synthetic class-dependent transition matrices used in our experiments on *MNIST*. The noise rate is set to 20%.

| Noise type | Sym. | | Asym. | | Pair. | | Trid. | | Ins. | |
|---|---|---|---|---|---|---|---|---|---|---|
| Method/Noise | 20% | 40% | 20% | 40% | 20% | 40% | 20% | 40% | 20% | 40% |
| NPCL | 98.66 | 98.21 | 98.89 | 96.14 | 98.06 | 97.50 | 98.84 | 97.62 | 97.25 | 95.75 |
| | ±0.03 | ±0.11 | ±0.02 | ±1.21 | ±0.24 | ±0.18 | ±0.05 | ±0.29 | ±0.63 | ±1.04 |
| INCV | 98.58 | 98.37 | 98.89 | 97.67 | 97.98 | 97.21 | 98.74 | 97.65 | 96.83 | 94.97 |
| | ±0.14 | ±0.06 | ±0.06 | ±0.26 | ±0.21 | ±0.39 | ±0.17 | ±0.32 | ±1.04 | ±1.35 |
| APL | 98.76 | 94.92 | 98.63 | 88.65 | 98.66 | 68.44 | 98.93 | 76.44 | 97.63 | 87.90 |
| | ±0.06 | ±0.31 | ±0.05 | ±1.72 | ±0.10 | ±2.95 | ±0.04 | ±3.04 | ±0.73 | ±1.94 |
| CDR | 94.77 | 92.16 | 96.73 | 91.05 | 93.25 | 71.02 | 94.06 | 70.28 | 93.17 | 77.45 |
| | ±0.17 | ±0.73 | ±0.19 | ±0.76 | ±0.90 | ±3.89 | ±0.92 | ±4.01 | ±0.96 | ±3.04 |

Table 7: Test accuracy (%) on *MNIST* over the last ten epochs.

• Asymmetic noise : this kind of label noise is generated by flipping labels within a set of similar classes. In this paper, for *MNIST*, flipping 2→7, 3→8, 5↔6. For *F-MNIST*, flipping TSHIRT→SHIRT, PULLOVER→COAT, SANDALS→SNEAKER. For *CIFAR-10*, flipping TRUCK→AUTOMOBILE, BIRD→AIRPLANE, DEER→HORSE, CAT↔DOG. For *CIFAR-100*, the 100 classes are grouped into 20 super-classes, and each has 5 sub-classes. Each class is then flipped into the next within the same super-class.

• Pairflip noise: the noise flips each class to its adjacent class.

• Tridiagonal noise: the noise corresponds to a spectral of classes where adjacent classes are easier to be mutually mislabeled, unlike the unidirectional pair flipping. It can be implemented by two consecutive pair flipping transformations in the opposite direction.

(2). Instance-dependent label noise:

• Instance noise: the noise is quite realistic, where the probability that an instance is mislabeled depends on its features. We generate this type of label noise to validate the effectiveness of the proposed method as did in (Xia et al., 2020; Yao et al., 2021).

We use synthetic noisy *MNIST* as an example and plot the noise transition matrices in Figure 4. The noise rate is set to 20%.

## B.2 COMPARISON WITH OTHER BASELINES

We focus on the sample selection approach in learning with noisy labels. In the main paper (Section 3.1), we fairly compare our methods with the baselines which also focus on sample selection. We add two other sample-selection baselines here. As suggested by the related survey (Song et al., 2020), the sample selection methods can be divided into three categories: (a) "Multi-network Learning", (b) "Multi-round Learning", and (c) "Hybrid Approach". As our method belongs to (a), we exploit INCV (Chen et al., 2019) which belongs to (b), and NPCL (Lyu & Tsang, 2020) which belongs to (c), to make the comparison more comprehensive.

Here, we also evaluate other types of baselines. We exploit APL (Ma et al., 2020) and CDR (Xia et al., 2021), which add implicit regularization from different perspectives. The experiments are conducted on *MNIST* and *F-MNIST*. Other experimental settings are the same as those in the main paper. The experimental results are provided in Tables 7 and 8, which show that the proposed methods are superior to these baselines with respect to classification performance.

| Noise type | Sym. | | Asym. | | Pair. | | Trid. | | Ins. | |
|---|---|---|---|---|---|---|---|---|---|---|
| Method/Noise | 20% | 40% | 20% | 40% | 20% | 40% | 20% | 40% | 20% | 40% |
| NPCL | 91.65 | 86.72 | 91.20 | 70.65 | 91.07 | 85.03 | 90.99 | 88.79 | 90.77 | 86.34 |
| | ±0.37 | ±2.30 | ±0.06 | ±2.83 | ±0.22 | ±1.94 | ±0.15 | ±1.04 | ±0.85 | ±2.78 |
| INCV | 91.66 | 85.39 | 91.35 | 65.82 | 90.26 | 85.73 | 91.76 | 88.42 | 89.54 | 86.21 |
| | ±0.31 | ±1.14 | ±0.70 | ±4.32 | ±0.38 | ±1.92 | ±0.17 | ±1.55 | ±1.79 | ±2.86 |
| APL | 91.73 | 89.06 | 90.13 | 80.34 | 90.22 | 78.54 | 90.84 | 86.53 | 90.96 | 85.55 |
| | ±0.20 | ±0.41 | ±0.17 | ±0.63 | ±0.80 | ±4.33 | ±0.22 | ±0.76 | ±0.77 | ±2.86 |
| CDR | 85.62 | 71.83 | 89.78 | 79.05 | 85.72 | 69.07 | 86.75 | 73.63 | 85.92 | 73.14 |
| | ±0.96 | ±1.37 | ±0.41 | ±1.39 | ±0.65 | ±2.31 | ±1.19 | ±2.82 | ±1.43 | ±3.12 |

Table 8: Test accuracy on *F-MNIST* over the last ten epochs.

## B.3 EXPERIMENTS ON SYNTHETIC *CIFAR-100*

For *CIFAR-100*, we use a 7-layer CNN structure from (Yu et al., 2019; Yao et al., 2020a). Other experimental settings are the same as those in the experiments on *MNIST*, *F-MNIST*, and *CIFAR-10*. The results are provided in Table 9. We can see the proposed method outperforms all the baselines.

| Noise type | Sym. | | Asym. | | Pair. | | Trid. | | Ins. | |
|---|---|---|---|---|---|---|---|---|---|---|
| Method/Noise | 20% | 40% | 20% | 40% | 20% | 40% | 20% | 40% | 20% | 40% |
| S2E | 44.59 | 25.78 | 42.18 | 26.81 | 42.99 | 26.96 | 43.16 | 27.72 | 43.13 | 27.12 |
| | ±0.32 | ±5.44 | ±1.73 | ±2.25 | ±1.54 | ±2.48 | ±0.93 | ±3.56 | ±0.67 | ±3.86 |
| MentorNet | 43.15 | 37.62 | 41.03 | 28.27 | 40.06 | 27.17 | 42.20 | 31.74 | 40.54 | 33.09 |
| | ±0.42 | ±0.89 | ±0.22 | ±0.41 | ±0.37 | ±0.92 | ±0.30 | ±0.88 | ±0.69 | ±1.53 |
| Co-teaching | 45.17 | 40.95 | 42.76 | 30.27 | 42.50 | 30.07 | 44.41 | 34.96 | 42.23 | 35.87 |
| | ±0.25 | ±0.52 | ±0.34 | ±0.33 | ±0.39 | ±0.17 | ±0.41 | ±0.35 | ±0.52 | ±1.47 |
| SIGUA | 42.03 | 40.53 | 36.67 | 26.71 | 36.48 | 26.73 | 39.21 | 32.69 | 39.19 | 33.51 |
| | ±0.33 | ±0.49 | ±0.25 | ±0.42 | ±0.37 | ±0.33 | ±0.40 | ±0.36 | ±0.32 | ±0.43 |
| JoCor | 45.93 | 41.56 | 42.89 | 29.19 | 42.12 | 30.12 | 44.98 | 34.23 | 44.28 | 35.60 |
| | ±0.21 | ±0.57 | ±0.37 | ±1.42 | ±0.35 | ±0.65 | ±0.27 | ±1.13 | ±0.59 | ±0.99 |
| CNLCU-S | **46.09** | **42.11** | **43.06** | **30.47** | **43.08** | **30.33** | **45.19** | **35.49** | **44.80** | **36.23** |
| | **±0.29** | **±0.70** | **±0.28** | **±0.37** | **±0.92** | **±0.74** | **±0.90** | **±1.30** | **±0.70** | **±0.49** |
| CNLCU-H | **46.27** | **42.05** | **43.21** | **30.55** | **43.25** | **30.79** | **45.02** | **35.24** | **45.02** | **36.17** |
| | **±0.38** | **±0.87** | **±0.93** | **±0.72** | **±0.75** | **±0.86** | **±1.06** | **±0.93** | **±1.07** | **±1.54** |

Table 9: Test accuracy (%) on *CIFAR-100* over the last ten epochs. The best two results are in bold.

## B.4 EXPERIMENTS FOR ABLATION STUDY

We conduct the ablation study to analyze the sensitivity of the length of time intervals. The results are shown in Figure 5 and 6. As we can seen, the proposed method, i.e., CNLCU-S and CNLCU-H are robust to the choices of hyperparameters.

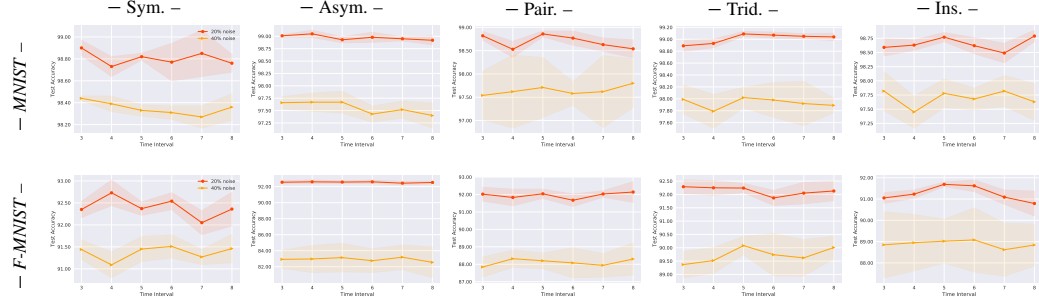

Figure 5: Illustrations of the hyperparameter sensitivity for the proposed CNLCU-S. The error bar for standard deviation in each figure has been shaded.

Note that in this paper, we concern uncertainty from *two aspects*, i.e., the uncertainty about small-loss examples and the uncertainty about large-loss examples. Here, we conduct ablation study to show the

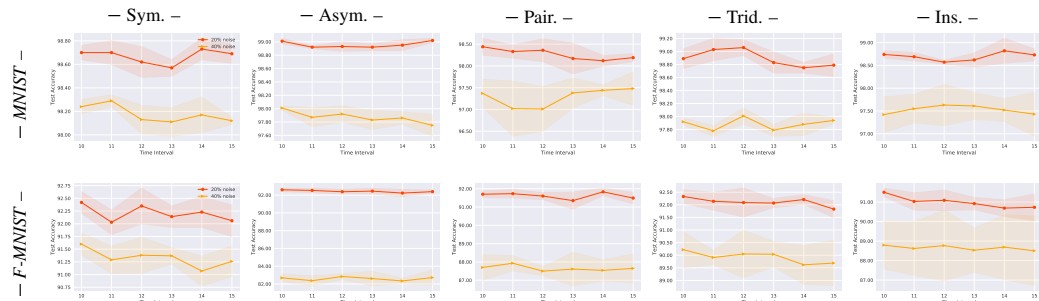

Figure 6: Illustrations of the hyperparameter sensitivity for the proposed CNLCU-H. The error bar for standard deviation in each figure has been shaded.

effect of removing different components to provide insights into what makes the proposed methods successful. The experiments are conducted on *MNIST* and *F-MNIST*. Other experimental settings are the same as those in the main paper (Section 3.1). Note that we employ two networks to teach each other following (Han et al., 2018). Therefore, when we do not consider uncertainty in sample selection, the proposed methods will *reduce to* the baseline Co-teaching (Han et al., 2018).

To study the effect of concerning uncertainty about small-loss examples, we remove the concerns about large-loss examples, i.e., the network is not encouraged to choose the less selected examples for updates. We express such a setting as "**w**ith**o**ut **c**oncerning about **l**arge-loss examples" (abbreviated as *w/o cl*). To study the effect of concerning uncertainty about large-loss examples, we remove the concerns about small-loss examples, i.e., we only exploit the predictions of the current network. We express such a setting as "**w**ith**o**ut **c**oncerning about **s**mall-loss examples" (abbreviated as *w/o cs*). Besides, we express the setting which directly uses non-robust mean as Co-teaching-M.

The experimental results of ablation study are provided in Tables 10 and 11. As can be seen, both aspects of uncertainty concerns can improve the robustness of models. Therefore, combining two uncertainty concerns, we can better combat noisy labels. In addition, robust mean estimation is superior to the non-robust mean in learning with noisy labels.

| Noise type | Sym. | | Asym. | | Pair. | | Trid. | | Ins. | |
|---|---|---|---|---|---|---|---|---|---|---|
| Method/Noise | 20% | 40% | 20% | 40% | 20% | 40% | 20% | 40% | 20% | 40% |
| CNLCU-S | 98.82 | 98.31 | 98.93 | 97.67 | 98.86 | 97.71 | 99.09 | 98.02 | 98.77 | 97.78 |
| | ±0.03 | ±0.05 | ±0.06 | ±0.22 | ±0.06 | ±0.64 | ±0.04 | ±0.17 | ±0.08 | ±0.25 |
| CNLCU-S *w/o cl* | 98.02 | 96.83 | 98.50 | 96.25 | 98.22 | 96.08 | 98.64 | 97.25 | 98.17 | 97.13 |
| | ±0.08 | ±0.29 | ±0.04 | ±0.13 | ±0.13 | ±0.75 | ±0.31 | ±0.24 | ±0.20 | ±0.40 |
| CNLCU-S *w/o cs* | 98.15 | 97.12 | 98.36 | 96.39 | 98.04 | 96.12 | 98.74 | 97.30 | 98.11 | 97.32 |
| | ±0.20 | ±0.22 | ±0.07 | ±0.48 | ±0.24 | ±0.68 | ±0.05 | ±0.52 | ±0.15 | ±0.43 |
| CNLCU-H | 98.70 | 98.24 | 99.01 | 98.01 | 98.44 | 97.37 | 98.89 | 97.92 | 98.74 | 97.42 |
| | ±0.06 | ±0.06 | ±0.04 | ±0.03 | ±0.19 | ±0.32 | ±0.15 | ±0.05 | ±0.16 | ±0.39 |
| CNLCU-H *w/o cl* | 98.06 | 96.92 | 98.39 | 96.51 | 97.04 | 95.62 | 98.33 | 97.41 | 98.01 | 96.15 |
| | ±0.13 | ±0.23 | ±0.04 | ±0.57 | ±0.87 | ±0.93 | ±0.47 | ±0.92 | ±0.20 | ±0.28 |
| CNLCU-H *w/o cs* | 98.19 | 97.05 | 98.76 | 97.17 | 97.26 | 96.31 | 98.29 | 97.65 | 98.34 | 96.49 |
| | ±0.22 | ±0.49 | ±0.59 | ±0.60 | ±1.19 | ±0.25 | ±0.17 | ±0.92 | ±0.36 | ±0.48 |
| Co-teaching-M | 97.72 | 97.78 | 98.27 | 95.42 | 96.22 | 95.01 | 97.92 | 96.64 | 98.02 | 96.03 |
| | ±0.08 | ±0.32 | ±0.03 | ±0.42 | ±0.10 | ±0.65 | ±0.14 | ±0.77 | ±0.04 | ±0.57 |
| Co-teaching | 97.53 | 95.62 | 98.25 | 95.08 | 96.05 | 94.16 | 98.05 | 96.18 | 97.96 | 95.02 |
| | ±0.12 | ±0.30 | ±0.08 | ±0.43 | ±0.96 | ±1.37 | ±0.06 | ±0.85 | ±0.09 | ±0.39 |

Table 10: Test accuracy (%) on *MNIST* over last ten epochs.

## B.5   Results on Synthetic Imbalanced Noisy Datasets

In the main paper, we have provided the test accuracy on synthetic imbalanced noisy datasets in Figure 2. Due to the limited space, we provide the selected ratio achieved on *IM-MNIST* and *IM-F-MNIST* in Table 12. The results explain well why our methods perform better on synthetic imbalanced noisy datasets, i.e., our methods can make better use of training examples with the imbalanced classes. Note that since we give a try to large-loss but less selected data in a conservative way, the selected

| Noise type | Sym. | | Asym. | | Pair. | | Trid. | | Ins. | |
|---|---|---|---|---|---|---|---|---|---|---|
| Method/Noise | 20% | 40% | 20% | 40% | 20% | 40% | 20% | 40% | 20% | 40% |
| CNLCU-S | 92.37 | 91.45 | 92.57 | 83.14 | 92.04 | 88.20 | 92.24 | 90.08 | 91.69 | 89.02 |
| | ±0.15 | ±0.28 | ±0.15 | ±1.77 | ±0.26 | ±0.44 | ±0.17 | ±0.34 | ±0.10 | ±1.02 |
| CNLCU-S *w/o cl* | 91.77 | 89.40 | 91.25 | 72.93 | 91.53 | 87.31 | 91.31 | 89.50 | 91.09 | 88.45 |
| | ±0.35 | ±0.26 | ±0.30 | ±2.63 | ±0.17 | ±0.59 | ±0.52 | ±0.32 | ±0.13 | ±0.57 |
| CNLCU-S *w/o cs* | 91.85 | 90.76 | 91.94 | 80.99 | 91.28 | 87.31 | 91.39 | 89.29 | 90.98 | 88.73 |
| | ±0.33 | ±0.28 | ±0.09 | ±2.74 | ±0.20 | ±0.72 | ±0.07 | ±0.51 | ±0.43 | ±0.62 |
| CNLCU-H | 92.42 | 91.60 | 92.60 | 82.69 | 91.70 | 87.70 | 92.33 | 90.22 | 91.50 | 88.79 |
| | ±0.21 | ±0.19 | ±0.18 | ±0.43 | ±0.18 | ±0.69 | ±0.26 | ±0.71 | ±0.21 | ±1.22 |
| CNLCU-H *w/o cl* | 91.70 | 90.05 | 91.08 | 71.35 | 91.03 | 87.22 | 91.59 | 90.01 | 90.80 | 88.31 |
| | ±0.04 | ±0.31 | ±0.06 | ±2.30 | ±0.29 | ±0.72 | ±0.07 | ±0.24 | ±0.27 | ±1.09 |
| CNLCU-H *w/o cs* | 91.82 | 90.92 | 92.45 | 80.73 | 91.21 | 87.49 | 92.08 | 89.72 | 91.21 | 88.62 |
| | ±0.13 | ±0.42 | ±0.25 | ±1.63 | ±0.17 | ±0.32 | ±0.13 | ±0.24 | ±0.38 | ±0.73 |
| Co-teaching-M | 91.33 | 89.05 | 91.14 | 71.03 | 90.85 | 86.95 | 91.50 | 89.18 | 90.74 | 88.25 |
| | ±0.18 | ±0.73 | ±0.90 | ±3.73 | ±0.61 | ±0.19 | ±0.46 | ±0.44 | ±1.06 | ±0.92 |
| Co-teaching | 91.48 | 88.80 | 91.03 | 68.07 | 90.77 | 86.91 | 91.24 | 89.18 | 90.60 | 87.90 |
| | ±0.10 | ±0.29 | ±0.14 | ±4.58 | ±0.23 | ±0.71 | ±0.11 | ±0.36 | ±0.12 | ±0.45 |

Table 11: Test accuracy (%) on *F-MNIST* over last ten epochs.

| Dataset | *IM-MNIST* | | | | *IM-F-MNIST* | | | |
|---|---|---|---|---|---|---|---|---|
| Method/Noise | 10% | 20% | 30% | 40% | 10% | 20% | 30% | 40% |
| S2E | 0.13 | 0.11 | 0.09 | 0.05 | 0.13 | 0.17 | 0.16 | 0.12 |
| | ±0.12 | ±0.05 | ±0.02 | ±0.01 | ±0.04 | ±0.03 | ±0.02 | ±0.04 |
| MentorNet | 0.10 | 0.15 | 0.12 | 0.13 | 0.12 | 0.15 | 0.09 | 0.14 |
| | ±0.02 | ±0.02 | ±0.03 | ±0.02 | ±0.01 | ±0.03 | ±0.01 | ±0.02 |
| Co-teaching | 0.09 | 0.07 | 0.05 | 0.12 | 0.17 | 0.04 | 0.13 | 0.07 |
| | ±0.03 | ±0.02 | ±0.01 | ±0.01 | ±0.05 | ±0.00 | ±0.04 | ±0.01 |
| SIGUA | 0.04 | 0.04 | 0.01 | 0.02 | 0.03 | 0.02 | 0.04 | 0.00 |
| | ±0.00 | ±0.00 | ±0.00 | ±0.00 | ±0.00 | ±0.00 | ±0.00 | ±0.00 |
| JoCor | 0.11 | 0.08 | 0.07 | 0.06 | 0.05 | 0.13 | 0.13 | 0.07 |
| | ±0.04 | ±0.01 | ±0.03 | ±0.02 | ±0.01 | ±0.04 | ±0.03 | ±0.02 |
| CNLCU-S | **0.60** | **0.37** | **0.39** | **0.38** | **0.35** | **0.39** | **0.36** | **0.30** |
| | **±0.11** | **±0.09** | **±0.04** | **±0.06** | **±0.03** | **±0.04** | **±0.03** | **±0.02** |
| CNLCU-H | **0.57** | **0.32** | **0.37** | **0.32** | **0.34** | **0.35** | **0.32** | **0.28** |
| | **±0.13** | **±0.01** | **±0.07** | **±0.05** | **±0.02** | **±0.06** | **±0.04** | **±0.03** |

Table 12: Selected ratio (%) on *IM-MNIST* and *IM-F-MNIST*. The best two results are in bold.

ratio is still far away from the class prior probability on the test set, i.e., 10%. However, a little improvement of the selection ratio can bring a considerable improvement of test accuracy. These results tell us that, in the sample selection approach when learning with noisy labels, improving the selected ratio of training examples with the imbalanced classes is challenging but promising for generalization. This practical problem deserves to be studied in depth.

### B.6 EXPERIMENTS WITH HIGH NOISE LEVELS

In the main paper, we set the noise rates to be smaller than 50%, which ensures that clean labels in noisy classes are diagonally dominant in all label noise settings. To demonstrate the effectiveness of our method with high noise levels, we conduct experiments on *MNIST* and *F-MNIST* with 50%, 60%, and 70% symmetric noise. The experimental results are presented in Table 13. As can be seen, the proposed methods can achieve the best performance in almost all cases.

## C COMPLEMENTARY EXPLANATION FOR NETWORK STRUCTURES

Table 14 describes the 9-layer CNN (Han et al., 2018) used on *MNIST*, *F-MNIST*, and *CIFAR-10*. Table 15 describes the 7-layer CNN (Yu et al., 2019) used on *CIFAR-100*. Here, LReLU stands for Leaky ReLU (Xu et al., 2015). The slopes of all LReLU functions in the networks are set to 0.01. Note that that the 7/9-layer CNN is a standard and common practice in weakly supervised learning. We decided to use these CNNs, since then the experimental results are directly comparable with previous approaches in the same area, i.e., learning with noisy labels.

| Dataset | MNIST | | | F-MNIST | | |
|---|---|---|---|---|---|---|
| Method/Noise | 50% | 60% | 70% | 50% | 60% | 70% |
| APL | 84.97±2.97 | 75.68±1.22 | 70.11±0.52 | 76.80±3.21 | 72.77±4.37 | 68.39±7.17 |
| CDR | 76.85±2.46 | 57.22±1.92 | 54.22±0.94 | 53.41±1.81 | 45.82±2.77 | 41.33±3.69 |
| INCV | 96.14±0.33 | 94.12±1.13 | 92.13±0.75 | 84.92±0.60 | 82.79±1.64 | 80.12±0.26 |
| NPCL | 97.05±0.14 | 94.77±0.73 | 93.58±1.26 | 86.07±1.33 | 83.29±2.65 | 79.39±1.58 |
| S2E | 87.76±2.39 | 79.24±4.79 | 50.30±7.82 | 62.20±3.94 | 59.88±1.27 | 45.88±7.96 |
| MentorNet | 91.14±0.17 | 90.11±0.37 | 88.72±0.46 | 86.51±0.11 | 85.91±0.44 | 83.27±0.55 |
| Co-teaching | 95.60±0.38 | 95.44±0.30 | 94.11±0.38 | 88.72±0.14 | 87.92±0.34 | 85.92±0.72 |
| SIGUA | 91.35±2.62 | 88.62±1.93 | 86.08±6.04 | 83.39±3.29 | 79.36±4.54 | 72.14±4.28 |
| JoCor | 97.14±0.10 | **96.47± 0.46** | 95.01±0.29 | 89.16±0.27 | 87.93±0.61 | 86.99±0.92 |
| CNLCU-S | **97.78±0.07** | **96.73±0.31** | **96.02±0.38** | **89.56±0.25** | **88.56±0.59** | **87.54±0.42** |
| CNLCU-H | **97.64±0.25** | 96.45±0.55 | **95.30±0.26** | **89.77±0.29** | **88.07±0.62** | **87.40±1.05** |

Table 13: Test accuracy on *MNIST* and *F-MNIST* with high noise levels over the last ten epochs. The best two results are in bold.

| CNN on *MNIST* | CNN on *F-MNIST* | CNN on *CIFAR-10* |
|---|---|---|
| 28×28 Gray Image | 28×28 Gray Image | 32×32 RGB Image |
| 3×3 conv, 128 LReLU | | |
| 3×3 conv, 128 LReLU | | |
| 3×3 conv, 128 LReLU | | |
| 2×2 max-pool | | |
| dropout, $p = 0.25$ | | |
| 3×3 conv, 256 LReLU | | |
| 3×3 conv, 256 LReLU | | |
| 3×3 conv, 256 LReLU | | |
| 2×2 max-pool | | |
| dropout, $p = 0.25$ | | |
| 3×3 conv, 512 LReLU | | |
| 3×3 conv, 256 LReLU | | |
| 3×3 conv, 128 LReLU | | |
| avg-pool | | |
| dense 128→10 | dense 128→10 | dense 128→10 |

Table 14: The CNN on *MNIST*, *F-MNIST*, and *CIFAR-10*.

| CNN on *CIFAR-100* |
|---|
| 32×32 RGB Image |
| 3×3 conv, 64 ReLU |
| 3×3 conv, 64 ReLU |
| 2×2 max-pool |
| 3×3 conv, 128 ReLU |
| 3×3 conv, 128 ReLU |
| 2×2 max-pool |
| 3×3 conv, 196 ReLU |
| 3×3 conv, 196 ReLU |
| 2×2 max-pool |
| dense 256→100 |

Table 15: The CNN on *CIFAR-100*.

## D NEEDED ASSUMPTIONS WITHOUT NOISY VALIDATION SETS

In the main paper, a noisy validation set is exploited to determine $\sigma^2$ and $\tau_{\min}$. We discuss, if we do not use the noisy validation set, what assumptions we will need.

- In statistical learning, it is an important problem for estimating the variance of a data distribution. However, it is hard to obtain an accurate estimation from a finite sample. To reduce the estimation errors, some kinds of assumptions are often needed, e.g., random sampling assumptions, replacement assumptions, and strict distribution assumptions. We suggest the readers to refer (Wolter, 2007) for more details.

- For the accurate estimation of $\tau_{\min}$, we have to assume that the Markov chain can exhibit a *cutoff*, which means that the total variation distance decreases very rapidly in a small interval (Paulin et al., 2015). Readers can check the Figure 1 of (Lubetzky & Sly, 2013) for better understanding.

## E FURTHER DISCUSSION ON LEARNING BY UNCERTAINTY

We discuss a related work named ActiveBias (Chang et al., 2017) on learning with uncertainty, although it studies the predicted probability, rather than the loss. In general, the differences between our work and ActiveBias lie in the different problems, different focuses, and different technical implementation. In more detail, we summarize the main differences between the two works in the following:

- ActiveBias mainly focuses on mining hard examples to help generalization. Although it mentions the label-noise problem, it does not consider comprehensive noise settings. Also,

the baselines are SGD, ADAM, and other optimization methods, but not the methods are specially designed for learning with noisy labels. In contrast, our work mainly focuses on learning with noisy labels and has more comprehensive experiments, which explore the performance of different label-noise methods under noisy supervision.

- ActiveBias focuses on using the hard examples, but not too difficult examples, which is stated in the original paper. In the imbalanced setting of this paper, clean imbalanced data are too difficult to distinguish with the predictions of models (Figure 1, right) and therefore are too difficult examples. Our methods aim to make use of such examples. Experiments show that such examples are of great importance for generalization.

- The introduction of uncertainty in ActiveBias relies on the estimation of the prediction variance but does not consider the bad influence of mislabeled data/outliers on this estimation. Our methods use the loss distribution and the side effect of mislabeled data is considered by robust mean estimators.

## F    RELATED WORK ON SAMPLE SELECTION

In this paper, we focus on the procedure of using the *small-loss* criterion for sample selection, which is very commonly used in learning with noisy labels and shared by the state-of-the-art methods. We review prior effects in this line. Existing methods (Jiang et al., 2018; Han et al., 2018; Yu et al., 2019; Wei et al., 2020; Lyu & Tsang, 2020) focus on the class-balanced problem. Based on the memorization effect of deep networks, they select examples with small losses for network updates and drop examples with large losses. The main reason is that clean examples are more likely to have small losses, but mislabeled examples are more likely to have large losses in class-balanced cases. Distinctly, these methods ignore the clean examples with large losses, following suboptimal generalization. When the training dataset is noisy and extremely imbalanced, prior methods are weak. It is because since they cannot select clean hard examples within non-dominant classes effectively based on losses, which is shown in the main paper. Unfortunately, such examples are critical for generalization. We are the first one to focus on this important problem and tackle it by considering the uncertainty of losses.

