# OpenReview forum: "Sample Selection with Uncertainty of Losses for Learning with Noisy Labels"
_ICLR.cc/2022/Conference — ICLR 2022 Poster_

### Official Review · Reviewer_7WR9 · 2021-10-28

**Correctness:** 3
**Technical Novelty And Significance:** 4
**Empirical Novelty And Significance:** 3
**Recommendation:** 6
**Confidence:** 4

**Main Review:**

Learning with noisy labels is an important issue and has attracted much attention. This paper proposes a novel sample selection method, namely CNLCU, to optimize model training. CNLCU is based on the probabilistic constraint relaxation and can select ‘good’ samples with time intervals.

Overall, the method proposed in the paper is technically sound. The authors provide complete derivations in the appendix, which shows the correctness of the features of the proposed method. The experimental design is reasonable, which follows the conventions in many noisy label learning studies. The comparison with well-selected baselines in different noise levels shows the advantages of the proposed method. The paper is well organized and clearly presented.

Weaknesses:
- It would better if the noise ratios cover a wider range rather than only 20% and 40% are selected. Usually, evaluations also should be made under heavily noisy situations where noise ratios are large than 50%. This will make the conclusion more convincing.
- The evaluation matric as well as the analysis should be diverse. CNLCU outperforms in terms of mean accuracy. E.g., if the dataset is imbalanced, the accuracy may be not enough to comprehensively describe the performance. I also notice that on some datasets, the performance of CNLCU appears larger variances, which should be further explained.
- This paper assumes that the training losses in L_t conform to a Markov process. This assumption may not be always true in reality. The position of W in parameter space depends on the starting points and search paths, but not merely the last positions. Some support evidences are better to be shown. Or, some statistical operations should be taken on the correlation of the W sequence during the training process.


**Summary Of The Paper:**

This paper proposes a probabilistic-based sample selection approach CNLCU to distinguish whether the training sample is mislabeled or ignored. The experimental results show that the proposed method outperforms most up-to-date sample selection methods.

**Summary Of The Review:**

Although some minor issues are better to be further addressed, CNLCU is indeed novel and technically sound. The experimental results show the potential superior performance of CNLCU. Thereby, I rate this paper with marginally above the acceptance threshold.

---

> ### Author Response · Authors · 2021-11-14
> **Response to Reviewer 7WR9**
>
> We thank the reviewer for the positive reviews. We provide our responses below.
>
> > **Q1:** The experiments with larger noise rates are suggested.
>
> **A1:** (1) To ensure that, in all noise settings, clean labels in noisy classes are diagonally dominant, we set the noise rates to be less than 50%.
> (2) We add experiments with 50%, 60%, and 70% symmetric noise in Appendix B.6 as suggested.
>
> > **Q2:** The evaluation matric as well as the analysis should be diverse.
>
> **A2:** Thanks for this nice comment. In Appendix B.5, we use the selected ratio of training examples in non-dominant classes to explain the experiments on class-imbalanced datasets. In the final version, we will add more discussions and analyses to comprehensively describe the performance.
>
> > **Q3:** On the assumption of the Markov process.
>
> **A3:** (1) We agree with you this assumption may not be always held. As in the process of network parameter updates, the parameters of the current model are mainly dependent on the parameters of the model at the last iteration, we approximate that the training losses conform to a Markov process. Comprehensive experimental results, especially for the results on real-world noisy datasets, justify the reasonability of this assumption.
> (2) For more evidence supporting our claims, we would add more mathematical analyses, e.g., from the view of network weight updates with different optimization methods.

---

> > ### Comment · Reviewer_7WR9 · 2021-11-21
> > **Response read**
> >
> > Thank the authors for providing their response to our concerns. Considering the overall quality of the paper, my recommendation score remains 6.

---

### Official Review · Reviewer_5G3L · 2021-11-01

**Correctness:** 4
**Technical Novelty And Significance:** 3
**Empirical Novelty And Significance:** 3
**Recommendation:** 8
**Confidence:** 3

**Main Review:**

Pros:
1. This paper is well written and easy to follow.
2. The goal of this paper is well-motivated, with a detailed discussion about the uncertainty of losses in sample selection.
3. The technical steps are clearly explained with theoretical justification.

Cons:

As shown in Table5, the difference between DivideMix-S/H and the original DivideMix is a little bit small. It will be more convincing if the authors could report the error bar of these experiments.


**Summary Of The Paper:**

This paper discusses the potential weaknesses in previous sample selection criteria in learning with noisy labels. And then propose a new selection criterion by incorporating the uncertainty of losses, together with theoretical justification. Experiments on both synthetic noisy balanced/imbalanced datasets and real-world noisy datasets validate the effectiveness of the proposed approach.

**Summary Of The Review:**

From my perspective, this work studies an important problem in previous sample selection criteria in learning with noisy labels. And the proposed method is clearly explained and theoretical analyses have been provided. Sufficient experiment results validate the effectiveness of this approach, although the results in semi-supervised learning seem not very convincing. Overall, I tend to accept this paper.


################# After rebuttal ##################\
Thanks for the responses. I still tend to accept this paper and keep my score.

---

> ### Author Response · Authors · 2021-11-14
> **Response to Reviewer 5G3L**
>
> We thank the reviewer for the positive reviews. We provide our responses below.
>
> > **Q1:** The error bars of the results in semi-supervised learning.
>
> **A1:** Due to the huge calculation cost of experiments on real-world noisy datasets, the experiments were conducted once in the main paper. Here, as suggested, to make results more convincing, we repeat the experiments five times. The experimental results are provided in Table 4-1.
>
> | Methods | Food-101 | WebVision (Mini)| Clothing1M |
> | :----: | :----: | :----: | :----: |
> | DivideMix | 86.71$\pm$0.06 | 77.36$\pm$0.13 | 74.60$\pm$0.18 |
> | DivideMix-S| 86.94$\pm$0.10 | 77.55$\pm$0.09 | 74.91$\pm$0.09 |
> | DivideMix-H| 86.86$\pm$0.09 | 77.50$\pm$0.06 | 74.80$\pm$0.13 |
>
> Table 4-1 Mean and standard deviations of test accuracy (%) on real-world noisy datasets.

---

### Official Review · Reviewer_LJ1a · 2021-11-01

**Correctness:** 3
**Technical Novelty And Significance:** 3
**Empirical Novelty And Significance:** 3
**Recommendation:** 6
**Confidence:** 4

**Main Review:**

The problem of label noise + (extreme) class imbalance is an interesting yet under studied direction with a potentially significant impact to many applications (e.g., worst-group generalization, fairness), which the paper does a good job brining attention to. The paper is well written and overall easy to follow, despite that some clarification may help readers better understand the proposed methods (see minor concern below).

Some of my major concerns are:

1. To implement the proposed method, it seems that naively one will need to keep track of the history of losses of every sample in the training set. For soft truncation (Eq. 7), one can simply keep a per-sample running mean and selection times in memory; for hard truncation (Eq. 8), it seems much more complicated. Can authors add some details of the space complexity overhead to use the proposed method? These can be argued as one major disadvantage when compared to other regularization-based LNL methods (Appendix B.2).
2. The hard truncation criterion is derived based on a Markovian assumption. Does it contradict with some popular tricks training DNN, such as temporal averaging, or momentum? For instance, it seems that all models in the paper are trained using ADAM where momentum is used hence the Markovian assumption is broken. Can the authors please comment?
3. The experiments mostly are designed with relatively low noise (20% and 40%). Sample selection can suffer typically from high-noise settings and the so-called self-confirmation trap. It would be good to compare the proposed method to other baselines in such settings, since so far all methods perform quite comparably across all datasets and noise types where the proposed method outperforms other slightly by 1-2% (Table 1, 2, 3, 9).

A minor concern/suggestion:

1. The paper only discussed related work very briefly in Sec 3.1 (Baselines). A dedicated related work section should help improve the paper: How do other sample selection consider different reasons for samples having large losses? Have they not considered at all? Any other paper that challenged the small-loss/large-loss assumption in any way?
2. How is Step 4/5 in Algorithm 1 optimized? At a first glance, this amounts to solving a combinatorial problem in each mini-batch? Perhaps the authors has a better way to do so?
3. How are hparams (sigma and tau_min) tuned given a *noisy* validation set?

**Summary Of The Paper:**

The paper proposes a novel sample selection method for learning with noisy labels (LNL). Based on the typical "small-loss" assumption, the motivation is to consider uncertainty about large-loss samples in order to distinguish the two confounding cases: truly mislabeled samples, or clean yet underrepresented samples that are less frequently selected or learned by the model so far. The proposed method explores robust mean estimation to summarize the per-sample loss trajectory, through soft truncation and hard truncation. Concentration inequalities are used to obtain the final selection criteria to perform conservative search. Results are overall promising, in particular showing very good results dealing with label noise + extreme class imbalance (Figure 2).

**Summary Of The Review:**

The proposed method is a novel method concerning an interesting yet under studied direction with a potentially significant impact to many applications (e.g., worst-group generalization, fairness), which the paper does a good job brining attention to. Results are overall promising, in particular showing very good results dealing with label noise + extreme class imbalance (Figure 2). Some concerns should be addressed during open discussion with authors.

---

> ### Author Response · Authors · 2021-11-14
> **Response to Reviewer LJ1a**
>
> We thank the reviewer for the positive reviews. We provide our responses below.
>
> > **Q1:** The discussions on the space complexity of the proposed method.
>
> **A1:** The space complexity overhead to use CNLCU-H is $n*(l_t+1)$ memory space that exploits to store the loss history and selected times of each example, where $n$ is the sample size and $l_t$ is the time-interval length (relatively small, e.g., 10~15). The space complexity is relatively low.
>
> > **Q2:** On the assumption of the Markov process.
>
> **A2:** (1) When the optimization method is SGD or Mini-batch SGD, the weights of a network in the current iteration depend on the weights in the last iteration.
> (2) When momentum is used in optimization, e.g., Adam, the weight updates still mainly rely on the weights in the last iteration. It is because the exponential average operation makes the weights in the last iteration contribute more to optimization. Therefore, we approximatively consider the training loss to conform to a Markov process.
> (3) A series of experimental results justify the reasonability of this assumption.
>
> > **Q3:** Experiments with larger noise rates.
>
> **A3:** We add experiments with 50%, 60%, and 70% symmetric noise in Appendix B.6 as suggested. The results verify the effectiveness of our method.
>
> > **Q4:** More discussions about related work on sample selection.
>
> **A4:** Thanks for the suggestions. We add discussions about prior sample selection methods based on the small loss criterion, which are provided in Appendix F.
>
> > **Q5:** How is Step 4/5 in Algorithm 1 optimized?
>
> **A5:** In steps 4 and 5, we select examples based on (7) or (8). This procedure is similar to the sample selection procedure in Co-teaching. However, we use different criteria that consider the uncertainty of losses.
>
> > **Q6:** How are hyperparameters tuned given a noisy validation set?
>
> **A6:** We tune hyperparameters according to the accuracy of the noisy validation set. The hyperparameters that can bring higher accuracy are used. The reason why the noisy validation set can work is that the clean labels are dominating in noisy classes and that noisy labels are random, the accuracy on the noisy validation set and the accuracy on the clean test data set are positively correlated.  Some methods on learning with noisy labels share a similar idea of using the noisy validation set, e.g., [1] and [2].
>
> [1] Duc Tam Nguyen et al. SELF: Learning to filter noisy labels with self-ensembling. ICLR 2020.
> [2] Xuefeng Li et al. Provably end-to-end label-noise learning without anchor points. ICML 2021.

---

> > ### Comment · Reviewer_LJ1a · 2021-11-29
> > **Thanks for the response!**
> >
> > Many thanks to the authors for the response. Those cleared my concerns.

---

### Official Review · Reviewer_45Q2 · 2021-11-02

**Correctness:** 4
**Technical Novelty And Significance:** 4
**Empirical Novelty And Significance:** 4
**Recommendation:** 8
**Confidence:** 2

**Main Review:**

Strengths
- The proposed approach is justified theoretically
- Many datasets and comparison methods used and thus the proposed approach is also validated empirically

Weaknesses
- I could not find a major weakness in the paper. It would be interesting to see the work extended to other types of data (NLP, Speech...)

Typos
- 2.3, Soft truncation : « given a random variable X » probably


**Summary Of The Paper:**

This paper proposes a novel algorithm to improve the sample selection in the case of noisy labels training by incorporating the uncertainty of losses. To select samples, the authors propose to use the lower bounds of the confidence intervals derived from concentration inequalities instead of using point estimation of losses. The authors introduce the Soft truncation estimator and the Hard truncation estimator and propose lower bounds for both estimators. The authors introduce an algorithm, the CNLCU Algorithm based on using either the lower bounds calculated for the Soft truncation estimator or the Hard truncation estimator.

The authors validate empirically their results on four benchmark datasets (MNIST, F-MNIST, CIFAR-10, CIFAR-100) and use a diverse set of possible noise functions. The method proposed almost always outperforms all other comparable approaches.


**Summary Of The Review:**

The authors propose a novel algorithm, justified theoretically and validated empirically to improve the sample selection in the case of noisy labels training.

---

> ### Author Response · Authors · 2021-11-14
> **Response to Reviewer 45Q2**
>
> We thank the reviewer for the positive reviews. We provide our responses below.
>
> > **Q1:** Typos.
>
> **A1:** We have corrected them after revision.
>
> > **Q2:** It would be interesting to see the work extended to other types of data.
>
> **A2:** Thanks for the suggestion. We extend this work to NLP data and exploit the NEWS dataset.
> We focus on the class-imbalanced cases. A 3-layer MLP in [1] is used. We exploit 10% and 40% asymmetric noise. Other settings follow the settings in the main paper. The results are presented in Table 2-1, which support our claims.
>
> [1] Songhua Wu et al. Class2simi: A noise reduction perspective on learning with noisy labels. ICML 2021.
>
> | Methods | S2E | MentorNet | Co-teaching | SIGUA | JoCor | CNLCU-S | CNLCU-H |
> | :----: | :----: | :----: | :----: | :----: | :----: | :----: | :----: |
> |10%| 52.75$\pm$0.84 | 52.06$\pm$0.27 | 52.05$\pm$0.32 | 41.38$\pm$0.67 | 52.01$\pm$0.24 | 54.12$\pm$0.17 | 54.08$\pm$0.28 |
> |40%| 43.84$\pm$1.62 | 47.88$\pm$1.84 | 49.33$\pm$1.13 | 32.49$\pm$0.85 | 48.77$\pm$0.12 | 50.98$\pm$0.65 | 50.77$\pm$0.54 |
>
> Table 2-1: Test accuracy (%) on IM-NEWS over the last ten epochs.

---

> > ### Comment · Reviewer_45Q2 · 2021-11-28
> > **Response Read**
> >
> > Thanks for the response and the additional experiments. I am satisfied with the authors' response and additional experiments. I keep my score the same (8).

---

### Official Review · Reviewer_jAuy · 2021-11-03

**Correctness:** 3
**Technical Novelty And Significance:** 3
**Empirical Novelty And Significance:** 3
**Recommendation:** 5
**Confidence:** 5

**Main Review:**

Strength: (1) Provide comprehensive analysis on one category of noisy label classification method. (2) experiment dataset is diverse covering many commonly used datasets and setups.

Weakness: (1) the improvement in classification performance is marginal. (2) A critical state-of-the-art is not provided (see below). (2) Experiment results could be improved (need a t-test proving the experiment results are significant). (3) The paper only considers one category of noisy-label classification methods. There are many other categories that may provide better performance than the proposed methods (e.g., early learning, error bounded approach, etc.).


**Summary Of The Paper:**

The paper performs a comprehensive analysis on a sub-category of noisy-label classification techniques. Following that, the paper proposed a new improved method to handle sample selection-based noisy-label classification techniques.

**Summary Of The Review:**

I like this paper. It discusses the weakness of some noisy-label learning methods. The paper also conducts a solid experiment (except missing one critical comparison -- MentorMix). MentorMix is also a sample selection-based approach. It has demonstrated superior performance in dealing with the noisy labels. Because MentorMix uses mixup to enrich data, it should be robust against data imbalance. I would like to see a comparison of MentorMix and the proposed method.

---

> ### Author Response · Authors · 2021-11-14
> **Response to Reviewer jAuy**
>
> Thanks for your comments. We address your concerns as follows.
>
> > **Q1:** The improvement is marginal. Experiment results could be improved.
>
> **A1:** For the experiments on simulated class-balanced noisy datasets, our method achieves the best performance in most cases. The improvement is also clear at some time, e.g., 20% and 40% asymmetric noise cases on F-MNIST and CIFAR-10. For the experiments on more challenging simulated class-imbalanced and real-world noisy datasets, the proposed method outperforms baselines clearly.
>
> > **Q2:** The paper only considers one category of noisy-label classification methods.
>
> **A2:** (1) As this paper studies an important problem in previous sample selection criteria in learning with noisy labels, we mainly employ the baselines that also work on sample selection, to support our claims. Besides, other types of baselines, e.g., robust loss functions and regularization, are exploited. The results are provided in Appendix B.2.
> (2) We understand your concern. Some advanced methods may bring better performance. However, the contribution of this paper is not limited to performance. This paper addresses weaknesses in previous sample selection methods and potentially impacts various applications (e.g., worst-group generalization and fairness). The contributions are sufficient, which are also mentioned by the other four reviewers.
>
> > **Q3:** Comparison with MentorMix.
>
> **A3:** (1) We agree with you that Mixup can relieve the issues of class imbalance. The main reason is that Mixup directly trains deep models on the entire dataset, rather than a subset of the dataset. This characteristic is fundamentally different from sample selection methods.
> (2) MentorMix is a sample-selection-based method. However, we must emphasize that MentorMix is a combination of two techniques, i.e., MentorNet (sample selection) and Mixup (regularization). By contrast, CNLCU only focuses on sample selection. Therefore, it is not very fair to compare our method with MentorMix.
> (3) To make a fair comparison, we boost our method by using Mixup. The experiments are conducted on IM-MNIST.  The experimental results are shown in Table 1-1, which verifies the effectiveness of our method.
>
> | Methods | 10% | 20% | 30% | 40% |
> | :----: | :----: | :----: | :----: | :----: |
> | MentorMix | 89.55$\pm$4.43 | 65.49$\pm$7.49 | 50.20$\pm$3.99 | 47.96$\pm$1.43 |
> | CNLCU-S+Mixup | 92.17$\pm$2.84 | 73.51$\pm$2.72 | 60.55$\pm$3.22 | 54.24$\pm$1.94 |
> | CNLCU-H+Mixup | 92.65$\pm$2.56 | 70.57$\pm$3.61 | 58.26$\pm$3.90 | 53.62$\pm$1.79 |
>
> Table 1-1: Comparison with MentorMix in test accuracy (%).

---

> ### Author Response · Authors · 2021-11-19
> **Response to Reviewer jAuy**
>
> Dear Reviewer jAuy:
>
> Thanks a lot for your efforts in reviewing this paper. We tried our best to address the mentioned concerns. Are there unclear explanations here? We could further clarify them.
>
> Best,
> Authors

---

> ### Author Response · Authors · 2021-11-28
> **Further Discussion**
>
> Dear Reviewer jAuy:
>
> Thanks again for your efforts in reviewing. We are still looking forward to your reply. Would you mind checking our response and confirming if there are unclear explanations?
>
> Best,
> Authors

---

### Author Response · Authors · 2021-11-14
**Summary of Revisions**

Dear reviewers and all,

We have uploaded a revised draft incorporating reviewer feedback. Below is a summary of the main changes: (1) Add experiments with high noise levels. (2) Add discussions on related work about sample selection. (3) Correct typos. We would highly appreciate it if you could read our responses and revisions. Please feel free to let us know if further details/explanations would be helpful.

Best,
Authors

---

### Public Comment · ~Deep_Patel2 · 2022-10-28
**Regarding proof of Theorem 1**

Hi,

I have the following doubt regarding the proof of Theorem 1 as proved in the paper. The analysis is essentially the same as that of [2] which, in turn, relies a lot on that in [1]. Although one could easily relax the need for identical distribution, one still needs the independence assumption (as discussed in Section 8 Pg. No. 32 in [1]) for the bounds to hold true. This still doesn't look like a reasonable assumption on the loss history (as maintained in this paper to compute the loss statistics) because the parameter updates are dependent on the previous iteration's parameters (values). So, the loss value is also naturally dependent on the previous iteration's parameters (values). This doesn't suggest independence between loss values of the sample for any pair of consecutive iterations.


References:

[1] Olivier Catoni. Challenging the empirical mean and empirical variance: a deviation study. In Annales de l’IHP Probabilités et statistiques, volume 48, pp. 1148–1185, 2012.

[2] Chen, Peng, Xinghu Jin, Xiang Li, and Lihu Xu. "A generalized Catoni’s M-estimator under finite α-th moment assumption with α∈(1, 2)." Electronic Journal of Statistics 15, no. 2 (2021): 5523-5544.

---

> ### Public Comment · ~Xiaobo_Xia1 · 2022-10-28
> **Response**
>
> Hi, thanks for pointing out this. By assuming that loss values are independent, Theorem 1 is established. As the assumption may be strong, and we do not introduce the assumption on the underlying distribution of losses, the uncertainty of losses can be changeable by tuning $\sigma$, which makes the uncertainty calculation adaptive for different cases.

---

> > ### Public Comment · ~Deep_Patel2 · 2022-10-29
> > **Can you elaborate a bit more?**
> >
> > Hi,
> >
> > Thanks for the speedy response. I think I didn't state my doubt clearly. Let me rephrase it like this: Will the bound -- as mathematically stated -- for Theorem 1 hold true even if the random variables are dependent? If so, could you please shed some light on how the proof of Theorem 1 could be modified to obtain the same or a similar mathematical expression for the bound?
> >
> > I understand that you are carefully tuning $\sigma$ and mixing time to take care of things empirically. But my doubt is whether this bound (Theorem 1) even holds true in the first place given that the random variables are dependent. As far as I understand, the independence assumption is critical; it's being used in the exponential moment inequalities in the very beginning of proofs of both [1] and [2] (cf. Proposition 2.1 on Page 5 in [1] or Lemma 2.1 on Page 6 in [2]).
> >
> > It would be really helpful if you could share a detailed comment about this.
> >
> >
> > References:
> >
> > [1] Olivier Catoni. Challenging the empirical mean and empirical variance: a deviation study. In Annales de l’IHP Probabilités et statistiques, volume 48, pp. 1148–1185, 2012.
> >
> > [2] Chen, Peng, Xinghu Jin, Xiang Li, and Lihu Xu. "A generalized Catoni’s M-estimator under finite α-th moment assumption with α∈(1, 2)." Electronic Journal of Statistics 15, no. 2 (2021): 5523-5544.

---

> > > ### Public Comment · ~Wang_Xinrui1 · 2023-04-08
> > > **Maybe such assumption is relatively mild**
> > >
> > > While it is true that the loss value in an iterative optimization algorithm is influenced by the previous iteration's parameters, if we consider these values over a longer time scale, they can be seen as random variables sampled from unknown distributions. Therefore, it can be difficult to determine whether the losses are dependent on the previous iteration or not, as the entire dataset contributes to the parameter updates in the model. It's the same dilemma as time series prediction. It's only my point of view about this iid assumption here. Overall, i think it can be strong sometimes but it's generally acceptable.

---

### Decision · Program_Chairs · 2022-01-20

**Decision:**

Accept (Poster)

**Comment:**

The manuscript discusses weaknesses in previous sample selection criteria in learning with noisy labels, and proposes a new selection criterion by incorporating the uncertainty of losses, together with theoretical justification. To select samples, the manuscript uses the lower bounds of the confidence intervals derived from concentration inequalities instead of using point estimation of losses. By incorporating uncertainty about large-loss samples, the method is able to distinguish: truly mislabeled samples, or clean yet underrepresented samples that are less frequently selected or learned by the model so far. Experiments are performed on four benchmark datasets (MNIST, F-MNIST, CIFAR-10, CIFAR-100) and use a diverse set of possible noise functions.

Reviewers agreed on several positive aspects of the manuscript, including:
1. This manuscript addresses weaknesses in previous sample selection methods and potentially impacts various applications (e.g., worst-group generalization and fairness);
2. The technical steps are clearly explained with theoretical justification;
3. Many datasets and comparison methods used and thus the proposed approach is also validated empirically.

Reviewers also highlighted several major concerns, including:
1. The space complexity of the proposed method; to implement the proposed method, it seems that naively one will need to keep track of the history of losses of every sample in the training set;
2. The validity of Markov process assumption on the training losses when momentum is used in optimization. Also, the position of the parameter in parameter space depends on the starting points and search paths, but not merely the last positions.
3. The experiments mostly are designed with relatively low noise (20% and 40%);
4. The evaluation matric as well as the analysis should be diverse. The performance of the proposed approach appears to have larger variances, which should be further explained.

Many of the major concerns have been addressed during the rebuttal including: experiments with 50%, 60%, and 70% symmetric noise, extension to NLP data by showing results on the NEWS dataset, and further discussion on the space complexity and Markov process assumption.

---

> ### Public Comment · ~Xinshao_Wang1 · 2022-07-08
> **An Open Question on whether clean or noisy validation set for ML/DL researchers caring about label noise**
>
> The authors mentioned: **We leave 10% noisy training data as a validation set for model selection.**
>
> However, please check the following discussions about **the flaw of this specific experimental setting**. To highlight, **I only would like to discuss this experimental setting, and I am not commenting on the technical quality of this paper.**
>
> * https://github.com/XinshaoAmosWang/Improving-Mean-Absolute-Error-against-CCE#open-reviews-and-discussion.
> * https://www.reddit.com/r/MachineLearning/comments/htgucz/r_when_talking_about_robustnessregularisation_our/?utm_source=share&utm_medium=web2x&context=3
> * https://www.reddit.com/r/MachineLearning/comments/hmt9ds/r_we_really_need_to_rethink_robust_losses_and/?utm_source=share&utm_medium=web2x&context=3
>
> ### Very nice discussions in reddit about "**A validation set serves a similar rule as a testing set. Both have to be clean.**"
> Link: https://www.reddit.com/r/MachineLearning/comments/nsh6uu/r_cvpr_2021progressive_self_label_correction/h10ga71/?utm_source=share&utm_medium=web2x&context=3
>
> Thanks.

---

> ### Public Comment · ~Gang_Niu1 · 2022-07-09
> **Re: An Open Question on whether clean or noisy validation set for ML/DL researchers caring about label noise**
>
> Thanks for your question!
>
> Let me first quote your question below.
>
> > Reviewer#3's opinion in final justification: `The validation sets are required to be clean, which greatly decrease the contribution. Many existing methods employ noisy validation set to choose hyper-parameters, e.g., when the risk is consistent. As minimizing risks on the noisy validation set is asymptotically equal to minimizing risk on the clean data.'
>
> > My opinion discussed with my collaborators: Following the ML literature, a validation set should be clean as we should not expect a ML model to predict noisy data well. In other words, we cannot evaluate/decide a model’s performance on noisy validation/test data. Our goal is to avoid learning faults from noisy data and generalise better during inference.
>
> Then please find my answer as follows.
>
> When noisy labels are IID in the training and validation sets and the label noise is CCN, the clean validation accuracy and the noisy validation accuracy have a linear relationship in expectation that **noisy_valid_acc = clean_valid_acc * (1 - noise_rate)**. Even though the noisy validation accuracy is **never an accurate estimate** to the clean validation accuracy at all as you wrote, **it is no problem to use it as the metric for the model and/or hyperparameter selection purposes**. That is why "as **minimizing** risks on the noisy validation set is **asymptotically** equal to **minimizing** risk on the clean data" as the reviewer of your paper wrote.
>
> There is a detailed discussion about the trick of using noisy validation data in learning with noisy labels: https://arxiv.org/abs/2012.04193. Indeed, under the optimistic assumption that the Bayes optimal classifier has a clean validation accuracy 1, the noisy validation accuracy of this Bayes optimal classifier is actually **1 - noise_rate**, according to Theorem 1 in this AAAI 2021 paper. Before the publication of this AAAI 2021 paper, this validation trick was empirically used for many years in this research area. So did the current paper.
>
> Hope my answer can clarify your confusion.

---

> > ### Public Comment · ~Xinshao_Wang1 · 2022-07-10
> > **Minimizing risks on the noisy validation set is  NOT asymptotically equal to minimizing risk on the clean data because the noise rate is usually unknown!**
> >
> > Dear Gang，
> >
> > Thanks so much for quoting my questions, and your detailed replies. It provides a lot of information and is fantastic for a further following discussion.
> >
> > I agree with what you said “**under the optimistic assumption that the Bayes optimal classifier has a clean validation accuracy $1$, the noisy validation accuracy of this Bayes optimal classifier is actually** $1 - noise\__rate$”.
> >
> > Consequently, minimizing risks on the noisy validation set is  **NOT** asymptotically equal to minimizing risk on the clean data as the noise rate is usually unknown! Therefore, my past reviewer was factually wrong due to the widely used wrong research legacy.
> > Even the noise rate is known, e.g., the noise rate of a noisy validation set is 20%, then an optimal classifier is supposed to produce 80% accuracy on it. But 80% does not necessarily mean an optimal classifier, some other non-optimal classifiers may also have 80% accuracy.
> >
> > I hope they’re clear and make sense. That is our responsibility to correct those wrong research legacy.
> >
> > Thanks. Kind regards.

---

> > > ### Public Comment · ~Gang_Niu1 · 2022-07-10
> > > **Never give up**
> > >
> > > It is your responsibility to correct those wrong research legacy. I am too busy, so I am sorry that I cannot help you on this.
> > >
> > > If you keep on going and never give up for more than 30 years, you will be extremely successful sooner or later.